# Locally private online change point detection

**Thomas Berrett**
Department of Statistics
University of Warwick
Coventry, CV4 7AL, U.K.
tom.berrett@warwick.ac.uk

**Yi Yu**
Department of Statistics
University of Warwick
Coventry, CV4 7AL, U.K.
yi.yu.2@warwick.ac.uk

## Abstract

We study online change point detection problems under the constraint of local differential privacy (LDP) where, in particular, the statistician does not have access to the raw data. As a concrete problem, we study a multivariate nonparametric regression problem. At each time point $t$, the raw data are assumed to be of the form $(X_t, Y_t)$, where $X_t$ is a $d$-dimensional feature vector and $Y_t$ is a response variable. Our primary aim is to detect changes in the regression function $m_t(x) = \mathbb{E}(Y_t | X_t = x)$ as soon as the change occurs. We provide algorithms which respect the LDP constraint, which control the false alarm probability, and which detect changes with a minimal (minimax rate-optimal) delay. To quantify the cost of privacy, we also present the optimal rate in the benchmark, non-private setting. These non-private results are also new to the literature and thus are interesting *per se*. In addition, we study the univariate mean online change point detection problem, under privacy constraints. This serves as the blueprint of studying more complicated private change point detection problems.

## 1 Introduction

Online change point detection has been an active statistical research area for decades, originated from the demand of a reliable quality control mechanism under time and resources constraints [e.g. 34]. In recent years, due to the advance of technology, the applications of online change point detection are well beyond quality control and include climatology, speech recognition, imaging processing, among many others. While collecting the data as they are generated, one wishes to detect the underlying distributional changes as soon as the changes occur.

As the ability of collecting and storing data improves exponentially, protecting users' privacy has become one of the central concerns of data science. In practice, many of the systems that we monitor contain sensitive information. For instance, change point algorithms are used in cyber security to detect attacks, with the ultimate aim often being to protect private information [24, 30, 1]. Other common application areas include public health [21, 17] and finance [18], in which many of the data involved are highly personal. Given the prevalence of such problems, we address two questions in this paper:

- whether we can detect changes without the need for direct access to the sensitive raw data;

- and what the cost of protecting privacy is in terms of the detection delay and accuracy.

Traditional anonymisation of data has been shown to be an outdated method of privacy protection, particularly in multivariate settings [28, 26]. Even when direct identifiers such as names and locations are removed, data sets often contain enough information for researchers to identify individual subjects, and hence there is a need to quantify and control the release of privileged information. Formal privacy constraints provide us with a rigorous framework in which we may tackle such problems and explore the fundamental limits of private methods. Constraints of this type have been imposed in analyses

35th Conference on Neural Information Processing Systems (NeurIPS 2021).

carried out by Apple [31, 29], Google [16] and Microsoft [10], and provides organisations with a way of demonstrating the General Data Protection Regulation compliance [7].

The most popular privacy constraint is that of differential privacy [14], which assumes the existence of a third party who can be trusted to handle all the raw data. In situations where this assumption cannot be made, which are the focus in our work, we strengthen this constraint and require our algorithms to satisfy local differential privacy [e.g. 20, 12], which, in particular, insists that the raw data are not accessed by anyone except its original holder. Solutions to a wide range of statistical problems have been given, and much underlying theory has been developed [e.g. 35, 12, 13, 27, 3, 15, 19].

In this paper, we are concerned with a multivariate nonparametric regression online change point detection problem, with privatised data. To be specific, we assume that the original data are $\{(X_i, Y_i)\}_{i \in \mathbb{N}^*} \subset \mathbb{R}^d \times \mathbb{R}$, where the regression functions

$$m_i(x) = \mathbb{E}(Y_i | X_i = x), \quad x \in \mathbb{R}^d, \quad i \in \mathbb{N}^*, \tag{1}$$

satisfy that $m_{i+1} \neq m_i$ if and only if $i = \Delta \in \mathbb{N}^*$. The goal is to find a stopping time $\widehat{t}$, which minimises the detection delay $(\widehat{t} - \Delta)_+$, while controlling the false alarm probability $\mathbb{P}(\widehat{t} < \Delta)$.

With the concern of maintaining privacy, we do not directly have access to the original data $\{(X_i, Y_i)\}_{i \in \mathbb{N}^*}$, but a privatised version. Specifically, the original data are transmitted through an $\alpha$-locally differentially private ($\alpha$-LDP) channel for some fixed $\alpha > 0$. The privatised data are denoted by $\{(W_i, Z_i)\}_{i \in \mathbb{N}^*} \subset \mathbb{R}^d \times \mathbb{R}$. In our upper bounds we restrict attention to non-interactive mechanisms [see e.g. 12], and so a privacy mechanism is given by a sequence $\{Q_i\}_{i \in \mathbb{N}^*}$ of conditional distributions, with the interpretation that $(W_i, Z_i)|(X_i, Y_i) = (x_i, y_i) \sim Q_i(\cdot|(x_i, y_i))$. For $\{Q_i\}_{i \in \mathbb{N}^*}$ to satisfy the $\alpha$-LDP constraint we require that

$$\sup_{i \in \mathbb{N}^*} \sup_{A} \sup_{(x,y),(x',y') \in \mathbb{R}^d \times \mathbb{R}} \frac{Q_i(A|(x,y))}{Q_i(A|(x',y'))} \leq e^\alpha. \tag{2}$$

In our lower bounds, however, we allow mechanisms to be sequentially interactive [e.g. 12], so that a privacy mechanism is given by $\{Q_i\}_{i \in \mathbb{N}^*}$, with the interpretation that

$$(W_i, Z_i)|\{(X_i, Y_i, W_{i-1}, Z_{i-1}, \ldots, W_1, Z_1) = (x_i, y_i, w_{i-1}, z_{i-1}, \ldots, w_1, z_1) \\ \sim Q_i(\cdot|(x_i, y_i, w_{i-1}, z_{i-1}, \ldots, w_1, z_1)).$$

Here the $\alpha$-LDP constraint requires that

$$\sup_{i \in \mathbb{N}^*} \sup_{A} \sup_{(x,y),(x',y') \in \mathbb{R}^d \times \mathbb{R}} \sup_{w_1, z_1, \ldots, w_{i-1}, z_{i-1}} \frac{Q_i(A|(x, y, w_{i-1}, z_{i-1}, \ldots, w_1, z_1))}{Q_i(A|(x', y', w_{i-1}, z_{i-1}, \ldots, w_1, z_1))} \leq e^\alpha.$$

Since our upper and lower bounds match, up to a logarithmic factor, we may conclude that simpler non-interactive procedures result in optimal performance for this problem.

We will assume throughout that $\alpha \leq 1$, though this can be relaxed to $\alpha \leq C$ for any $C > 0$. This restricts attention to the strongest constraints, and is often the regime of primary interest [e.g. 12].

## 1.1 Summary of contributions and related literature

To the best of our knowledge, this is the first work on a few fronts.

Firstly, this is the first paper to consider change point detection under local privacy constraints. Previous work has focused on the central model of differential privacy, where there exists a third party trusted to have access to all of the data. [9] use established tools from the central model of differential privacy to detect changes in both the offline and online settings. The pre- and post-change distributions are assumed to be known, and a private version of the likelihood ratio statistic is analysed. Further development of these ideas, in particular the extension to detection of multiple changes in the online setting, is given in [39]. [6] give differentially private tests of simple hypotheses, shown to be optimal up to constants, which are then applied to the change point detection problem with known pre- and post-change distributions. In a setting in which the distributions are unknown, [8] develop private versions of the Mann–Whitney test to detect a change in location. The problem has also been studied under different notions of central privacy [23].

Secondly, this is the first paper to study the fundamental limits in multivariate nonparametric regression change point detection problems. We have derived the minimax rate of the detection delay,

allowing the jump size $\|m_\Delta - m_{\Delta+1}\|_\infty$, the variance of the additive noise $\sigma^2$ and the privacy constraint $\alpha$ to vary with the location of the change point $\Delta$. There has been a vast body of literature discussing the detection boundary and optimal estimation in the offline change point analysis [e.g. 33, 36]. Their counterparts in online change point analysis are relatively scarce and existing work includes univariate mean change [e.g. 37] and dynamic networks [e.g. 38]. On a separate note, multivariate nonparametric regression estimation, under privacy constraints, is studied in [4], and classification is studied in [2]

In addition, we have also provided the analysis and results based on the univariate mean online change point detection problem, with privatised data. This is, arguably, the simplest privatised, online change point detection problem. The analysis and results we shown in this paper enrich statisticians' toolboxes and serve as a benchmark for more complex problems.

## 2 Methodology

In this section, we describe our private change point detection algorithm, which takes the privatised data as input. The whole algorithm consists of two key ingredients: (1) the privacy mechanism and (2) the change point detection method.

**The privacy mechanism.** Throughout this paper, a binned estimator is the core of the analysis. Recall that the raw data at time point $i$ include a $d$-dimensional feature vector $X_i$, which we assume is supported within some bounded set $\mathcal{X} \subset \mathbb{R}^d$, and a univariate response variable $Y_i$. We denote $\{A_{h,j}\}_{j=1,\ldots,N_h}$ as a set of cubes of volume $h^d$, such that $\{\mathcal{X} \cap A_{h,j}\}_{j=1,\ldots,N_h}$ is a partition of $\mathcal{X}$, and write $x_{h,j}$ for the centre of $A_{h,j}$. The data point $(X_i, Y_i)$ is then randomised by taking

$$W_{i,j} = \mathbb{1}_{\{X_i \in A_{h,j}\}} + 4\alpha^{-1}\epsilon_{i,j} \quad \text{and} \quad Z_{i,j} = [Y_i]_{-M}^M \mathbb{1}_{\{X_i \in A_{h,j}\}} + 4M\alpha^{-1}\zeta_{i,j},$$

where $\{\epsilon_{i,j}, \zeta_{i,j}\}$ are independent and identically distributed standard Laplace random variables, and where $[Y]_{-M}^M = \min(M, \max(Y, -M))$ with $M > 0$ a truncation parameter. It is shown in Proposition 1 in [4] [see also 2] that this non-interactive mechanism is an $\alpha$-LDP channel. We emphasise that privacy is guaranteed without any assumptions on the distribution of $(X, Y)$.

**The change point detection method.** Given data $\{(W_{i,j}, Z_{i,j})\}$, as for online change point detection, we propose Algorithm 1, with the CUSUM estimator defined in Definition 1 and the nonparametric estimators involved defined in Definition 2.

---

**Algorithm 1** Online change point detection via CUSUM statistics

---

**INPUT:** $\{(W_{u,j}, Z_{u,j})\}_{u,j=1,2,\ldots} \subset \mathbb{R}^d \times \mathbb{R}$, $\{b_{s,t}, 1 \le s < t < \infty, \ldots\} \subset \mathbb{R}$.
  $t \leftarrow 1$, FLAG $\leftarrow 0$;
  **while** FLAG $= 0$ **do**
    $t \leftarrow t+1$; FLAG $= 1 - \prod_{s=1}^{t-1} \mathbb{1}\left\{\widehat{D}_{s,t} \le b_{s,t}\right\}$;
  **end while**
**OUTPUT:** $t$.

---

**Definition 1.** *Given a sequence $\{(W_{t,j}, Z_{t,j})\}$ and a pair of integers $1 \le s < t$, we define the CUSUM statistic*

$$\widehat{D}_{s,t} = \max_{j=1}^{N_h} \sqrt{s(t-s)/t} \left|\widehat{m}_{1:s}(x_{h,j}) - \widehat{m}_{(s+1):t}(x_{h,j})\right|,$$

*where $\widehat{m}_{\cdot:\cdot}(\cdot)$ is defined in Definition 2.*

**Definition 2.** *Given a sequence $\{(W_{t,j}, Z_{t,j})\}$ and a pair of integers $1 \le s < t$, with $h > 0$ being our bandwidth parameter, we define the regression function estimator as*

$$\widehat{m}_{s:t}(x) = \frac{\widehat{\nu}_{s:t}(A_{h,j})}{\widehat{\mu}_{s:t}(A_{h,j})} \mathbb{1}_{\left\{\widehat{\mu}_{s:t}(A_{h,j}) \ge \frac{\log(t-s+2)}{t-s+1}\right\}}, \quad \text{if } x \in A_{h,j},$$

*where*

$$\widehat{\nu}_{s:t}(A_{h,j}) = (t-s+1)^{-1} \sum_{s \le i \le t} Z_{i,j} \quad \text{and} \quad \widehat{\mu}_{s:t}(A_{h,j}) = (t-s+1)^{-1} \sum_{s \le i \le t} W_{i,j}.$$

Algorithm 1 is a standard online change point detection procedure. Whenever a new data point stamped at $t$ is collected, one checks if a change point has occurred by checking if any of $(\widehat{D}_{s,t})_{s=1,\ldots,t-1}$ have exceeded pre-specified thresholds. Apparently, both the storage and computation costs of this procedure are high. As pointed out in [37], without knowing the pre- and/or post-change point distributions, there is no existing algorithm which can have a constant computational cost $O(1)$ when a new data point is collected. Of course, one can also be smarter: instead of scanning through $s \in \{1, \ldots, t-1\}$, one can just scan through a dyadic grid as described in [38].

The CUSUM statistic $\widehat{D}_{s,t}$, defined in Definition 1, is a normalised difference between two estimated functions, before and after time point $s$, evaluated at each cell in the partition. It is used to examine, given all the data up to $t$, if $s$ is a possible change point. It can be viewed as a sample version of the quantity

$$\max_{s=1}^{t-1} \sqrt{\frac{s(t-s)}{t}} \left\| s^{-1} \sum_{i=1}^{s} m_i - (t-s)^{-1} \sum_{i=s+1}^{t} m_i \right\|_\infty,$$

where $\| \cdot \|_\infty$ is the sup-norm of a function. In [25], a similar CUSUM was proposed in studying the nonparametric density change point detection problem. In our setting, we are interested in change points in the regression functions. The construction of estimators is detailed in Definition 2. This estimator has previously studied in the one sample estimation scenario in [4], where it was shown to be a universally strongly consistent estimator of the true regression function.

## 3 Theory

In this section, we present our core results. All the assumptions are collected in Section 3.1. The theoretical guarantees of Algorithm 1 are presented in Section 3.2, which also includes a minimax lower bound result showing our proposed method is optimal, off by at most a logarithmic factor. To quantify the cost of maintaining $\alpha$-LDP, we also provide a benchmark result of the same data generating mechanism but without privacy in Section 3.3.

### 3.1 Assumptions

**Assumption 1** (Setup). *Assume that $\{(X_i, Y_i)\}_{i \in \mathbb{N}^*}$ is a sequence of independent random objects, taking values in $\mathcal{X} \times \mathbb{R}$, such that $\{X_i\}_{i \in \mathbb{N}^*}$ are independent each with distribution $\mu$ on $\mathcal{X}$. We assume that there exists an absolute constant $c_{\min} > 0$ such that $\mu(A_{h,j}) \geq c_{\min} h^d$ for all $A_{h,j}$ in the partition of $\mathcal{X}$. Assume that the regression functions $m_i(\cdot)$, given by (1), are well defined for $\mu$-almost all $x$ and $i \in \mathbb{N}^*$, such that there exists an absolute constant $C_{\mathrm{Lip}} > 0$ with*

$$\sup_{i \in \mathbb{N}^*} |m_i(x_1) - m_i(x_2)| \leq C_{\mathrm{Lip}} \|x_1 - x_2\|, \quad \forall x_1, x_2 \in \mathcal{X}.$$

*We also assume that there exists $\sigma > 0$ such that for all $\lambda \in \mathbb{R}$ we have*

$$\sup_{i \in \mathbb{N}^*} \sup_{x \in \mathbb{R}^d} \mathbb{E}\left( e^{\lambda \{Y_i - m_i(x)\}} \mid X_i = x \right) \leq e^{\frac{\lambda^2 \sigma^2}{2}}$$

*and that $\sup_{i \in \mathbb{N}^*} \sup_{x \in \mathbb{R}^d} |m_i(x)| \leq M_0$ for some $M_0 > 0$.*

Assumption 1 is the main model assumption. It is a general feature of the local differential privacy constraint that it is not possible to work with unbounded parameter spaces [see e.g. 11, Appendix G]. Thus, we assume that the feature vectors are supported in a bounded set $\mathcal{X}$, and we moreover assume that the regression functions $m_i$ are bounded, though these bounds are allowed to vary with the pre-change sample size $\Delta$. In addition, we also assume that the regression functions are Lipschitz with a constant $C_{\mathrm{Lip}}$. The Lipschitz condition can be easily relaxed to other type of continuity conditions, e.g. Hölder continuity. We assume the additive noise is sub-Gaussian with parameter $\sigma$, which is allowed to vary with the pre-change sample size. Assumption 1 is required to ensure that our algorithm controls the false alarm probability correctly and to ensure its optimality.

**Assumption 2** (No change point). *Assume that $m_1 = m_2 = \cdots$.*

**Assumption 3** (One change point). *Assume that there exists a positive integer $\Delta \geq 1$ such that*

$$m_1 = \cdots = m_\Delta \neq m_{\Delta+1} = m_{\Delta+2} = \cdots.$$

*In addition, let $\kappa = \|m_\Delta - m_{\Delta+1}\|_\infty$.*

**Assumption 4** (Signal-to-noise ratio). *There exists a sufficiently large absolute constant $C_{\mathrm{SNR}} > 0$ such that*

$$\kappa^2 h^{2d}\alpha^2\Delta/\max\{\sigma^2,\, M_0^2\} \geq C_{\mathrm{SNR}}\log\left\{\Delta/(c_{\min}h^{2d}\gamma)\right\},$$

*where $\gamma \in (0,1)$ is the desired bound on false alarm probability and $h$ is the bin-width used in constructing the estimators.*

Assumptions 2, 3 and 4 describe different scenarios and aspects of the change point assumptions. Assumption 2 is a formal assumption describing when there is no change point and Assumption 3 depicts the scenario when there is one change point. Recall that in Definition 1 we proposed the CUSUM statistic which is a sample version of a normalised sup-norm difference between two functions. This is to be consistent with $\kappa$, the characterisation of the jump, introduced in Assumption 3.

Assumption 4 is the signal-to-noise ratio condition, and is required when we study the optimality of our algorithm to detect changes. Recall that $\kappa$ is the jump size, $\sigma$ is the fluctuation size, $M_0$ is the upper bound of the mean function $m_i$, $\alpha$ is the privacy constraint and $\Delta$ is the size of the pre-change point sample size. Assumption 4 requires that $\kappa^2 h^{2d}\alpha^2\Delta\min\{M_0^{-2},\sigma^{-2}\}$ is larger than a logarithmic factor. This in fact allows $\kappa$, $\sigma$, $M_0$ and $\alpha$ to vary as $\Delta$ diverges. It may appear to be unnatural to involve tuning parameters in the signal-to-noise ratio condition. We remark that the involvement of $h$ in Assumption 4 is to provide more flexibility in the tuning parameter selection. We will elaborate this point in Section 3.2.

### 3.2 Optimal online change point detection with privatised data

Theorem 1 below is the main result, which shows that for any $\gamma \in (0,1)$, with properly chosen tuning parameters, with probability at least of $1-\gamma$, Algorithm 1 does not have false alarms and has a detection delay upper bounded by $\epsilon$ in (12).

**Theorem 1.** *Consider the settings described in Assumption 1. Let $\gamma \in (0,1)$ and $\widehat{t}$ be the stopping time returned by Algorithm 1 with inputs $\{(W_{tj}, Z_{tj})\}_{t,j=1,2,\ldots}$ and $\{b_{s,t}\}_{t=2,3,\ldots;s=1,\ldots,t}$, where*

$$b_{s,t} = \begin{cases} 2\sqrt{\dfrac{s(t-s)}{t}}\Big\{2(M-M_0)e^{-\frac{(M-M_0)^2}{2\sigma^2}} \\ \quad + C_{\mathrm{Lip}}\sqrt{d}h\Big\} + \dfrac{M}{c_{\min}h^d\alpha}\sqrt{\log\left(\dfrac{72t^3}{\gamma c_{\min}h^d}\right)}, & \text{if } \dfrac{s(t-s)}{t}c_{\min}^2 h^{2d}\alpha^2 \geq 64\log\left(\dfrac{72t^3}{\gamma c_{\min}h^d}\right); \\ \\ \infty, & \text{otherwise.} \end{cases}$$

$$(3)$$

*Assume that the truncation parameter satisfies*

$$M \geq M_1 = M_0 + \sigma\sqrt{2\log(2+\sigma/h) + \log\log(2+\sigma/h)} \tag{4}$$

*and the bandwidth satisfies $h \leq C\kappa$, where $C > 0$ is an absolute constant.*

*If Assumption 2 holds, then*

$$\mathbb{P}_\infty\left\{\widehat{t} < \infty\right\} < \gamma. \tag{5}$$

*Under Assumption 3, we have*

$$\mathbb{P}_\Delta\left\{\widehat{t} \leq \Delta\right\} < \gamma, \tag{6}$$

*for any $\Delta \geq 1$. If Assumptions 3 and 4 both hold, then*

$$\mathbb{P}_\Delta\left\{\Delta < \widehat{t} \leq \Delta + \epsilon\right\} \geq 1-\gamma, \quad \text{where} \quad \epsilon = C_\varepsilon\frac{M^2}{\kappa^2 h^{2d}\alpha^2}\log\left(\frac{\Delta}{h^{2d}c_{\min}\gamma}\right) \tag{7}$$

*and $C_\varepsilon > 0$ depends only on $d$.*

To better understand Theorem 1, we first inspect the sources of the error in the procedure. The estimators of the regression functions $m_i$ are defined in Definition 2, which is a binned estimator averaging over cubes of volume $h^d$. This is a typical nonparametric estimator, which brings in both bias and variance. On top of this, due to the constraints of privacy, we truncate the responses by $M$ in

the privacy channel. The truncation level $M$ should be an upper bound on $M_1$ – a large-probability upper bound on the response, consisting of the upper bound on the regression function and the additive noise – so that the truncation bias is no larger than the bias due to smoothing. On the other hand, larger values of $M$ result in larger variance, due to the need to add more noise in the privacy mechanism. If $M < M_1$ then the change point may be undetectable, as the bias could be larger than the signal. The same phenomenon occurs in nonparametric testing and change point detection problems, that the smoothing parameter should not be too large to mask the signal.

Algorithm 1 declares the existence of a change point $t$, if there exists an integer pair $(s, t)$ such that $\widehat{D}_{s,t} > b_{s,t}$. The threshold sequence is detailed in (3). It is separated into two cases: (i) when $s(t - s)/t$ is large enough for both $\widehat{m}_{1:s}$ and $\widehat{m}_{(s+1):t}$ to be estimated accurately, and (ii) otherwise. In view of the sources of errors, we can see that in case (i), with probability at least $1 - \gamma$, the threshold $b_{s,t}$ is set to be the sum of an upper bound of all sources of errors. In case (ii), the threshold is set to be infinity, so that we never declare a change.

When there is no change point, or when there is a change point but $t \leq \Delta$, the thresholds, with probability at least $1 - \gamma$, are upper bounds on the estimators. When there is a change point and $t > \Delta$, due to Assumption 4, one can always let $s = \Delta$ such that there are enough samples to provide a good estimator of $m_\Delta$, the pre-change regression function. If $t - \Delta$ is not large enough such that case (i) holds, then we cannot provide a good estimator of $m_{\Delta+1}$. Once enough data are collected after the change point, Algorithm 1 is able to tell the difference and declare a change point, with delay upper bounded by $\epsilon$ in (12). Note that the conditions of case (i) will be satisfied before time $\Delta + \epsilon$.

We require the bandwidth $h \leq C\kappa$ to ensure that the binning will not smooth out the jump, and this condition is necessary. In practice, the bin-width can be chosen in a data-driven way [e.g. 38]. In view of Assumption 4, the smaller $h$ is, the larger $\kappa^2\alpha^2\Delta/\max\{\sigma^2, M_0^2\}$ needs to be. In the best case that $h \asymp \kappa$, Assumption 4 and $\epsilon$ in (12) read as

$$\frac{\kappa^{2+2d}\alpha^2\Delta}{\max\{\sigma^2, M_0^2\}} \gtrsim \log\left(\frac{\Delta}{c_{\min}\kappa^{2d}\gamma}\right) \quad \text{and} \quad \epsilon = C_\varepsilon \frac{M^2}{\kappa^{2+2d}\alpha^2}\log\left(\frac{\Delta}{\kappa^{2d}c_{\min}\gamma}\right). \tag{8}$$

We remark that the detection delay in (12) is of order

$$\epsilon \asymp \frac{M^2}{\kappa^2 h^{2d}\alpha^2}\log(\Delta/(h^{2d}\gamma)) \asymp \frac{(M - M_0)^2 + M_0^2 + \sigma^2}{\kappa^2 h^{2d}\alpha^2}\log(\Delta/(h^{2d}\gamma)),$$

which again reflects all three sources of errors. Now the question is whether this rate can be improved. To answer the question, we consider a simplified scenario, where we assume that $M_0 \leq \sigma$.

**Theorem 2** (Lower bound). *Denote by $\mathcal{P}_{\kappa,\sigma,\Delta}$ the class of distributions satisfying Assumptions 1 and 3, and assume $M_0 \leq \sigma$. Given $\alpha > 0$ let $\mathcal{Q}_\alpha$ be the collection of all sequentially interactive $\alpha$-locally differentially private mechanisms. Given $\gamma > 0$ consider the class of change point estimators*

$$\mathcal{D}(\gamma) = \big\{T : T \text{ is a stopping time wrt. the natural filtration and satisfies } \mathbb{P}_\infty(T < \infty) \leq \gamma\big\}.$$

*Then for sufficiently small $\gamma$, it holds that*

$$\inf_{Q \in \mathcal{Q}_\alpha} \inf_{\widehat{t} \in \mathcal{D}(\gamma)} \sup_{P \in \mathcal{P}_{\kappa,\sigma,\Delta}} 2\kappa^{2+2d}\sigma^{-2}\alpha^2\mathbb{E}_P\big\{(\widehat{t} - \Delta)_+\big\} \geq \log(1/\gamma).$$

Theorem 2 studies the private minimax rate of the detection delay in the framework proposed in [12]. Compared with standard minimax theory, new tools are required in order to deal with an arbitrary privacy mechanism in $\mathcal{Q}_\alpha$. We use Lemma 1 in [12, Supplementary material], which provides a uniform bound on the log-likelihood ratio of two distributions seen through a private channel. To the best of our knowledge, this is the first time that these tools have been applied to change point problems. Although, in our upper bounds, we only had to consider non-interactive mechanisms, this lower bound also applies to general sequentially interactive mechanisms. In particular, this shows that the use of interactive mechanisms is unnecessary in our problem.

Since in the minimax sense the lower bound is taken to be the infimum over all possible estimators, to make the results comparable in Assumption 4 and Theorem 1, we let $h \asymp \kappa$ and compare Theorem 2 with (8). Theorem 2 shows that the lower bound on the detection delay is of order $\sigma^2/(\kappa^{2+2d}\alpha^2)\log(1/\gamma)$, which compared to (8) is off by a logarithmic factor and therefore shows that Algorithm 1 is nearly minimax rate optimal.

We now provide a sketch of the proof of Theorem 2, focusing on the non-interactive case for simplicity. The proof is conducted on an online change point detection framework used in [22] and [37], based on a change of measure argument. The LDP ingredient used in this proof is devised by [12].

To be specific, we first construct a pair of distributions based on Lipschitz regression functions $f_1$ and $f_2$ on some ball $B(0, r_{\mathcal{X}})$, where $r_{\mathcal{X}} = O(1)$, $f_1 \equiv 0$ and $f_2(x) = (\kappa - \|x\|)\mathbb{1}_{\{x \in B(0,\kappa)\}}$. Note that, when $\kappa$ is small, the regression functions only differ in a small region, and thus this is a difficult change to detect. Given $f_1$ and $f_2$, the two distributions of $(X, Y)$ are denoted $P_1$ and $P_2$, where $X$ has the same marginal distribution under both while $Y|X = x \sim \text{Unif}[f_k(x) - \sigma, f_k(x) + \sigma]$ under $P_k$ for $k = 1, 2$. Letting $Q$ be any $\alpha$-LDP non-interactive privacy mechanism, write $P^n_{\kappa, \sigma, \nu}$ for the joint distribution of the first $n$ privatised data points when $(X_i, Y_i) \sim P_1$ for $i \leq \nu$ and $(X_i, Y_i) \sim P_2$ for $i > \nu$. By controlling the size of the log-likelihood ratio $Z_{\nu, n} = \log(dP^n_{\kappa, \sigma, \nu}/dP^n_{\kappa, \sigma, \infty})$ we show that any algorithm that has probability of false alarm bounded by $\gamma$ will require at least $\sigma^2 \log(1/\gamma)/(\kappa^{2+2d}\alpha^2)$ observations after the change point to reliably detect the change from $P_1$ to $P_2$ at time $\nu$.

As each observation is independent, we may decompose $Z_{\nu, n} = \sum_{i=\nu+1}^{n} Z_i$. A straightforward calculation shows that $d_{\text{TV}}(P_1, P_2) \lesssim \kappa^{d+1}/\sigma$, and then ideas from [12] are applied to see that $|Z_i| \lesssim \alpha\kappa^{d+1}/\sigma$ and $\mathbb{E}Z_i \lesssim \alpha^2\kappa^{2d+2}/\sigma^2$ for any $\alpha$-LDP privacy mechanism. We use the Azuma–Hoeffding inequality to extend this to show that

$$\mathbb{P}_{\kappa, \sigma, \nu}\left\{ \max_{1 \leq t \leq \frac{\sigma^2 \log(1/\gamma)}{\kappa^{2+2d}\alpha^2}} Z_{\nu, \nu+t} \geq \frac{3}{4}\log(1/\gamma) \;\middle|\; W_1, \ldots, W_\nu \right\} \leq \gamma \quad \text{a.s.}$$

Using the change of measure argument [e.g. 22, 37], the above bound leads us to complete the proof.

## 3.3 Optimal online change point detection with non-private data

As a benchmark, we provide the non-private counterpart of Theorem 1 and 2 in this subsection. For completeness, we also detail the counterparts of Definitions 1, 2 and Algorithm 1 in Definitions 3, 4 and Algorithm 2, respectively. We will conclude this subsection with comparisons of results in the private and non-private cases, quantifying the cost of maintaining privacy.

**Definition 3.** *Given a sequence $\{(X_t, Y_t)\}_{t \in \mathbb{N}^*}$ and a pair of integers $1 \leq s < t$, we define the CUSUM statistic*
$$\widetilde{D}_{s,t} = \max_{i=1}^{t} \sqrt{s(t-s)/t}\left|\widetilde{m}_{1:s}(X_i) - \widetilde{m}_{(s+1):t}(X_i)\right|,$$
*where $\widetilde{m}_{\cdot:\cdot}(\cdot)$ is defined in Definition 4.*

**Definition 4.** *Given a sequence $\{(X_t, Y_t)\}_{t \in \mathbb{N}^*}$, a pair of integers $1 \leq s < t$ and a tuning parameter $h > 0$, we define the regression function estimator as*
$$\widetilde{m}_{s:t}(x) = \frac{\nu_{s:t}(A_{h,j})}{\mu_{s:t}(A_{h,j})}, \quad \text{if } x \in A_{h,j},$$
*where*
$$\nu_{s:t}(A_{h,j}) = (t-s+1)^{-1}\sum_{s \leq i \leq t} Y_i\mathbb{1}_{\{X_i \in A_{h,j}\}} \quad \text{and} \quad \mu_{s:t}(A_{h,j}) = (t-s+1)^{-1}\sum_{s \leq i \leq t} \mathbb{1}_{\{X_i \in A_{h,j}\}}.$$

---

**Algorithm 2** Online change point detection via CUSUM statistics

---

**INPUT:** $\{(X_u, Y_u)\}_{u=1,2,\ldots} \subset \mathbb{R}^p \times \mathbb{R}, \{\tilde{b}_{u,t}, t = 2, 3, \ldots; u = 1, \ldots, t\} \subset \mathbb{R}$.
  $t \leftarrow 1; \text{FLAG} \leftarrow 0$;
  **while** $\text{FLAG} = 0$ **do**
    $t \leftarrow t + 1; \text{FLAG} = 1 - \prod_{s=1}^{t-1} \mathbb{1}\left\{\widetilde{D}_{s,t} \leq \tilde{b}_{s,t}\right\}$;
  **end while**
**OUTPUT:** $t$.

---

**Assumption 5** (Non-private signal-to-noise ratio)**.** *There exists a sufficiently large absolute constant $C_{\text{SNR}} > 0$ such that*
$$\kappa^2 h^d \Delta \sigma^{-2} \geq C_{\text{SNR}} \log(\Delta/(\gamma h^d)).$$

The following two theorems are the non-private version counterparts of Theorems 1 and 2. Theorem 4 shows that the detection delay rate we obtain in Theorem 3 is optimal off by a logarithmic factor.

**Theorem 3.** *Consider the settings described in Assumption 1. Let $\gamma \in (0,1)$ and $\widetilde{t}$ be the stopping time returned by Algorithm 2 with inputs $\{(X_t, Y_t)\}_{t=1,2,\dots}$ and $\{\widetilde{b}_{s,t}\}_{t=2,3,\dots;s=1,\dots,t}$, where*

$$\widetilde{b}_{s,t} = 2\sqrt{\frac{s(t-s)}{t}} C_{\mathrm{Lip}} \sqrt{d} h + \frac{4\sigma}{\sqrt{c_{\min} h^d}} \sqrt{5\log(t) + \log(32/\gamma)} \tag{9}$$

*Assume that the bandwidth satisfies $h \leq C\kappa$.*

*If Assumption 2 holds, then*

$$\mathbb{P}_\infty \left\{ \widetilde{t} < \infty \right\} < \gamma. \tag{10}$$

*Under Assumption 3 we have*

$$\mathbb{P}_\Delta \left\{ \widetilde{t} \leq \Delta \right\} < \gamma, \tag{11}$$

*for any $\Delta \geq 1$. If Assumptions 3 and 5 both hold, then*

$$\mathbb{P}_\Delta \left\{ \Delta < \widetilde{t} \leq \Delta + \epsilon \right\} \geq 1 - \gamma - (c_{\min} h^d)^{-1} \exp(-\Delta c_{\min} h^d), \tag{12}$$

*where*

$$\epsilon = C_\varepsilon \frac{\sigma^2}{\kappa^2 h^d} \log(\Delta/\gamma),$$

*with $C_\varepsilon > 0$ depending only on $d$.*

**Theorem 4.** *Denote by $\mathcal{P}_{\kappa,\sigma,\Delta}$ the class of distributions satisfying Assumptions 1 and 3 For any $\gamma \in (0,1)$, consider the class of change point estimators*

$$\mathcal{D}(\gamma) = \left\{ T : T \text{ is a stopping time wrt. the natural filtration and satisfies } \mathbb{P}_\infty(T < \infty) \leq \gamma \right\}.$$

*Then for any sufficiently small $\gamma$, it holds that*

$$\inf_{\widehat{t} \in \mathcal{D}(\gamma)} \sup_{P \in \mathcal{P}_{\kappa,\sigma,\Delta}} 2\kappa^{2+d} \sigma^{-2} \mathbb{E}_P \left\{ (\widehat{t} - \Delta)_+ \right\} \geq \log(1/\gamma).$$

We are now ready to quantify the cost of maintaining the privacy in the multivariate nonparametric regression online change point detection scenario.

• As we have pointed out, in order to maintain the privacy, we require $\mathcal{X}$ to be a bounded set, the regression function to be upper bounded by $M_0$ and an extra tuning parameter $M$ is introduced in truncation. All these are not needed in the non-private case.

• A more prominent difference roots in the detection delay rate. In the non-private case, the denominator is $\kappa^{2+d}$, which roots in the optimal nonparametric estimation [e.g. 32]. In the private case, the corresponding rate is $\kappa^{2+2d}$. This is in line with the literature in local differential privacy, where the curse of dimensionality is typically worse in nonparametric and high-dimensional problems than in their non-private counterparts [12, 2, 5, 3].

• Another difference is in terms of the privacy parameter $\alpha$, which only shows up in the private case. Recalling that we restrict ourselves to the most interesting regime $\alpha \in (0, 1]$, the effect of $\alpha$ is twofold: (1) Comparing Assumptions 4 and 5, we see that the private case requires a larger signal by a factor of $\alpha^{-2}$; and (2) comparing the detection delay rates in Theorem 1 and 3, we see that the rate in the private case is also inflated by a factor of $\alpha^{-2}$.

## 4  Numerical study

In this section we present the results of a numerical study of our locally private method's performance. We consider raw data $(X_1, Y_1), \dots, (X_n, Y_n)$ with $n = 10000, \Delta = 5000, X_i \sim \mathrm{Unif}[0,1]$ and $Y_i \sim \mathrm{Unif}[m_i(x) - 1/2, m_i(x) + 1/2]$, where

$$m_\Delta \equiv 0 \quad \text{and} \quad m_{\Delta+1}(x) = (1/2)\min(1, \max(5 - 10x, -1)).$$

To speed up computation, we modify Algorithm 1 so that $\widehat{D}_{s,t}$ is only calculated and compared to its threshold when $t \in \{n/100, 2n/100, \dots, n\}$. We privatise the data using $\alpha$ in the range

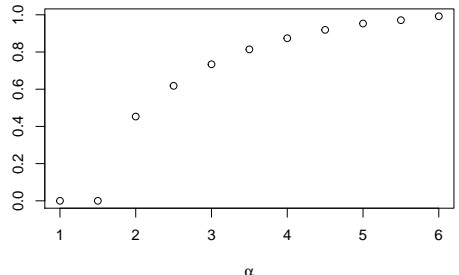 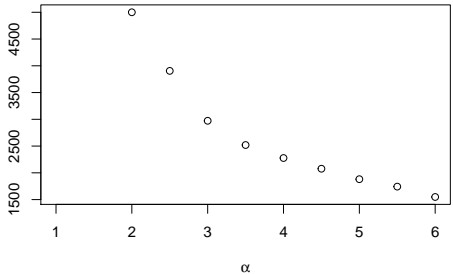

Figure 1: Proportion of trials with a change flagged    Figure 2: Average detection delay given a flag

$\{1, 1.5, 2, \ldots, 6\}$. For each value of $\alpha$, we assume that we have a privatised sample of size $n$ from the pre-change distribution, here $\text{Unif}[0, 1] \times \text{Unif}[-1/2, 1/2]$. The choices $M = 1$ and $h = 0.2$ are used to privatise the data and calculate the test statistics. We permute this sample $B = 1000$ times to choose our thresholds, as follows: for a range of values of $C$ we run the modified Algorithm 1 on each permutation of the privatised data with the choice

$$
b_{s,t} = \begin{cases} \dfrac{C}{h\alpha}\sqrt{\log\left(\dfrac{t}{\gamma h}\right)} & \text{if } \frac{s(t-s)}{t}h^2\alpha^2 \geq C^2 \log\left(\dfrac{t}{\gamma h}\right); \\[2em] \infty, & \text{otherwise} \end{cases}
$$

and we choose the minimal value of $C$ for which the overall false alarm probability is bounded above by $\gamma = 0.1$. With the thresholds chosen we ran the experiment over 1000 repetitions, and results are presented in Figures 1 and 2. The false detection probability was bounded by $\gamma$ for all values of $\alpha$. Full details of the implementation and simulation study can be found in the code available online.

## 5   Conclusions

We studied a multivariate, nonparametric regression, online, privatised change point detection problem. The method we proposed is shown to be minimax optimal in terms of its detection delay, with a theory-guided tuning parameter. As a benchmark result, we have also provided its counterpart for non-private data. The comparisons enable us to understand the cost of maintaining privacy.

In addition to the main results in the paper, in Appendix A we investigate an online univariate mean change point detection problem with privatised data. It includes a minimax lower bound on the detection delay and a polynomial-time algorithm which provides a matching upper bound, saving for a logarithmic factor. The framework we set up in Appendix A serves as a blueprint to study private online change detection with more complex data types.

Comparing the results in Appendix A to those in the non-private setting [e.g. 37] and comparing these differences with those we examined at the end of Section 3.3, one can see that we pay different costs for privacy in different data types. This leads to our future work, studying private change point detection in high-dimensional, functional or other nonparametric data, and understanding the tradeoff between accuracy and privacy in these more challenging situations.

Regarding the setup we have in this paper, it can be easily adjusted to allow for multiple change points. This is because, with large probability, our algorithm will not declare false alarms and will correctly detect a change point with a delay $\epsilon$ being a small fraction of $\Delta$. Therefore, provided that two consecutive change points are at least $\Delta$ apart, refreshing the algorithm whenever a change point is declared can enable us to detect multiple change points accurately.

Another interesting but challenge future work direction is to allow temporal dependence, say weakly dependent time series. It is not even clear how to define valid privacy mechanisms in this setting: if the time points were strongly dependent then our mechanisms would give more information as time progressed. In fact, this is even an open problem without the presence of change points. We would also need to adjust the concentration inequalities we are currently using to those which are suitable for weakly dependent data.

## Supplementary material

The supplementary material contains all the technical details and code of this paper.

## Acknowledgments and Disclosure of Funding

Funding in direct support of this work: DMS-EPSRC EP/V013432/1.

## A    A privatised univariate mean online change point detection

This is a self-contained section and the notation used in this section is independent from the notation in the rest of the paper. We start by considering a simple Laplace privacy mechanism and CUSUM statistics.

**Assumption 6.** *Assume the privatised data are $\{Z_1, Z_2, \ldots\}$ satisfying $Z_i = X_i + \epsilon_i/\alpha$, where (i) $\{X_1, X_2, \ldots\}$ is the original data and is a sequence of independent random variables with unknown means $\mathbb{E}(X_i) = f_i$, $i = 1, 2, \ldots$ and such that $\sup_{i=1,2,\ldots} \|X_i\|_{\psi_2} \leq \sigma$; (ii) $\{\epsilon_1, \epsilon_2, \ldots\}$ is a sequence of independent and identically distributed standard Laplace random variables, i.e. each $\epsilon_t$ has density $z \mapsto \exp(-|z|)/2$ on $\mathbb{R}$; and (iii) $\alpha > 0$ is a pre-specified value.*

Note that when $X_i$ takes values in an interval of length one $[a, a+1]$ then the privacy mechanism given by $Z_i$ is $\alpha$-LDP. This can be generalised to intervals of any fixed length by increasing the scale of the Laplace noise appropriately. To extend to unbounded variables we may truncate as in our main methodology in Section 2.

**Assumption 7.** *Assume that there exists a positive integer $\Delta \geq 1$ such that $f_1 = \cdots = f_\Delta \neq f_{\Delta+1} = f_{\Delta+2} = \cdots$. Let $\kappa = |f_\Delta - f_{\Delta+1}|$.*

**Assumption 8.** *There exists a sufficiently large absolute constant $C_{\mathrm{SNR}} > 0$ such that $\Delta \kappa^2 (\sigma^2 + 4\alpha^{-2})^{-1} \geq C_{\mathrm{SNR}} \log(\Delta/\gamma)$.*

We consider the CUSUM statistics $\widehat{D}_{s,t} = |\{(t-s/(ts)\}^{1/2} \sum_{l=1}^{s} Z_l - [s/\{t(t-s)\}]^{1/2} \sum_{l=s+1}^{t} Z_l|$ and run Algorithm 1.

**Theorem 5.** *Consider the settings described in Assumption 6. Let $\gamma \in (0, 1)$ and $\widehat{t}$ be the stopping time returned by Algorithm 1 with inputs $\{Z_t\}_{t=1,2,\ldots}$ and $\{b_t\}_{t=2,3,\ldots}$, where*

$$b_t = 2^{3/2}\sqrt{\sigma^2 + 4\alpha^{-2}} \log^{1/2}(t/\gamma). \tag{13}$$

*If $\Delta = \infty$, then*

$$\mathbb{P}_\infty \left\{ \widehat{t} < \infty \right\} < \gamma. \tag{14}$$

*Under Assumption 7, we have*

$$\mathbb{P}_\Delta \left\{ \widehat{t} \leq \Delta \right\} < \gamma, \tag{15}$$

*for any $\Delta \geq 1$. If Assumptions 7 and 8 both hold, then*

$$\mathbb{P}_\Delta \left\{ \Delta < \widehat{t} \leq \Delta + C_d \frac{(\sigma^2 + 4\alpha^{-2}) \log(\Delta/\gamma)}{\kappa^2} \right\} \geq 1 - \gamma, \tag{16}$$

*where $C_d > 0$ depends only on $d$ and satisfies $C_d < C_{\mathrm{SNR}}$.*

**Theorem 6.** *Denote $\mathcal{P}_{\kappa,\sigma,\Delta}$ be the class of distributions satisfying Assumptions 6 and 7 supported on an interval of fixed finite length. Denote $\mathcal{Q}_\alpha$ be the collection of all sequentially interactive $\alpha$-locally differentially private mechanisms, $\alpha > 0$. Consider the class of change point estimators*

$$\mathcal{D}(\gamma) = \big\{ T : T \text{ is a stopping time with respect to the natural filtration}$$

$$\text{and satisfies } \mathbb{P}_\infty(T < \infty) \leq \gamma \big\}.$$

*Then for small enough $\gamma$, and for $\kappa, \sigma > 0$ with $\kappa < 2\sigma$ and $\kappa + 2\sigma < 1$, it holds that*

$$\inf_{Q \in \mathcal{Q}_\alpha} \inf_{\widehat{t} \in \mathcal{D}(\gamma)} \sup_{P \in \mathcal{P}_{\kappa,\sigma,\Delta}} \mathbb{E}_P \left\{ (\widehat{t} - \Delta)_+ \right\} \geq \frac{c\sigma^2 \alpha^{-2} \log(1/\gamma)}{\kappa^2},$$

*where $c > 0$ is an absolute constant.*

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
