# Locally private online change point detection
# (Supplementary materials)

**Thomas Berrett**
Department of Statistics
University of Warwick
Coventry, CV4 7AL, U.K.
`tom.berrett@warwick.ac.uk`

**Yi Yu**
Department of Statistics
University of Warwick
Coventry, CV4 7AL, U.K.
`yi.yu.2@warwick.ac.uk`

This file consists of all the technical details needed in the main text.

## S.1 Proofs of main results

We first introduce some additional notation. For any integer pair $1 \leq s < t$, denote the un-smoothed and smoothed population CUSUM statistics as

$$D_{s,t} = \sup_{x \in \mathbb{R}^p} D_{s,t}(x) = \sqrt{\frac{s(t-s)}{t}} \sup_{x \in \mathbb{R}^p} |m_{1:s}(x) - m_{(s+1):t}(x)| \tag{S.1}$$

and

$$D_{s,t}^{(h)} = \sup_{x \in \mathbb{R}^p} D_{s,t}^{(h)}(x) = \sqrt{\frac{s(t-s)}{t}} \sup_{x \in \mathbb{R}^p} |m_{1:s}^{(h)}(x) - m_{(s+1):t}^{(h)}(x)|,$$

where

$$m_{s:t}(x) = \frac{1}{t-s+1} \sum_{i=s}^{t} m_i(x)$$

and

$$m_{s:t}^{(h)}(x) = \frac{1}{t-s+1} \sum_{i=s}^{t} \mathbb{E}\{Y_i | X_i \in A_{h,j}\}.$$

*Proof of Theorem 3.* For any integer pair $(s,t)$, with $1 \leq s < t$, note that

$$\left| \widetilde{D}_{s,t} - D_{s,t} \right| = \left| \sqrt{\frac{s(t-s)}{t}} \max_{i=1}^{t} \left| \widetilde{m}_{1:s}(X_i) - \widetilde{m}_{(s+1):t}(X_i) \right| - D_{s,t} \right|$$

$$\leq \sqrt{\frac{s(t-s)}{t}} \left| \max_{i=1}^{t} |\widetilde{m}_{1:s}(X_i) - \widetilde{m}_{(s+1):t}(X_i)| - \max_{i=1}^{t} |m_{1:s}^{(h)}(X_i) - m_{(s+1):t}^{(h)}(X_i)| \right|$$

$$+ \sqrt{\frac{s(t-s)}{t}} \left| \max_{i=1}^{t} |m_{1:s}^{(h)}(X_i) - m_{(s+1):t}^{(h)}(X_i)| - \sup_{x \in \mathbb{R}^p} |m_{1:s}^{(h)}(x) - m_{(s+1):t}^{(h)}(x)| \right|$$

$$+ \left| \sqrt{\frac{s(t-s)}{t}} \sup_{x \in \mathbb{R}^p} |m_{1:s}^{(h)}(x) - m_{(s+1):t}^{(h)}(x)| - D_{s,t} \right|$$

$$= (I) + (II) + (III). \tag{S.2}$$

**Step 1.** In order to show (10) and (11), we focus on the integer pairs $(s,t)$, with $1 \leq s < t \leq \Delta \leq \infty$. In this case, it follows from Lemma S.1 that $D_{s,t} = 0$, from Lemma S.2 that $(II) = 0$, from

Lemma S.3 that $(III) = 0$, from Lemma S.5 that with probability $1 - \gamma$,

$$(I) \leq 2\sqrt{\frac{s(t-s)}{t}}C_{\text{Lip}}\sqrt{d}h + \frac{4\sigma}{\sqrt{c_{\min}h^d}}\sqrt{5\log(t) + \log(16/\gamma)}.$$

Therefore, the results (10) and (11) hold due to the choice of $b_{s,t}$ in (9).

**Step 2.** In order to show (12), due to (S.2), it suffices to show that

$$D_{\Delta,\Delta+\epsilon} - (I) - (II) - (III) > \widetilde{b}_{\Delta,\Delta+\epsilon}$$

with probability at least $1 - \gamma$. In this case, it follows from Lemma S.1 that

$$D_{\Delta,\Delta+\epsilon} = \kappa\sqrt{\Delta}\sqrt{\frac{\epsilon}{\Delta+\epsilon}},$$

from Lemma S.2 that

$$\mathbb{P}\left\{(II) = 0\right\} \geq 1 - \frac{1}{c_{\min}h^d}\exp(-\Delta c_{\min}h^d),$$

for $C_{\text{SNR}}$ large enough, from Lemma S.3 that

$$(III) \leq 2\sqrt{\frac{\Delta\epsilon}{\Delta+\epsilon}}C_{\text{Lip}}\sqrt{d}h$$

and from Lemma S.5 that

$$(I) \leq 2\sqrt{\frac{\Delta\epsilon}{\Delta+\epsilon}}C_{\text{Lip}}\sqrt{d}h + \frac{4\sigma}{\sqrt{c_{\min}h^d}}\sqrt{5\log(\Delta+\epsilon) + \log(32/\gamma)}$$

with probability at least $1 - \gamma/2$. Note that, due to Assumption 5, $\Delta \geq \epsilon$. Combining the above statements, we thus have with probability at least $1 - \gamma$ that

$$D_{\Delta,\Delta+\epsilon} - (I) - (II) - (III) - \widetilde{b}_{\Delta,\Delta+\epsilon}$$

$$\geq \kappa\sqrt{\Delta}\sqrt{\frac{\epsilon}{\Delta+\epsilon}} - 2\sqrt{\frac{\Delta\epsilon}{\Delta+\epsilon}}C_{\text{Lip}}\sqrt{d}h - \frac{4\sigma}{\sqrt{c_{\min}h^d}}\sqrt{5\log(\Delta+\epsilon) + \log(32/\gamma)}$$

$$- 2\sqrt{\frac{\Delta\epsilon}{\Delta+\epsilon}}C_{\text{Lip}}\sqrt{d}h - 2\sqrt{\frac{\Delta\epsilon}{\Delta+\epsilon}}C_{\text{Lip}}\sqrt{d}h - \frac{4\sigma}{\sqrt{c_{\min}h^d}}\sqrt{5\log(\Delta+\epsilon) + \log(32/\gamma)}$$

$$\geq \kappa\sqrt{\Delta}\sqrt{\frac{\epsilon}{\Delta+\epsilon}} - 6\sqrt{\frac{\Delta\epsilon}{\Delta+\epsilon}}C_{\text{Lip}}\sqrt{d}h - \frac{8\sigma}{\sqrt{c_{\min}h^d}}\sqrt{5\log(64\Delta/\gamma)} > 0,$$

where the second inequality follows from $\epsilon \leq \Delta$, and the last inequality holds with a large enough $C_{\text{SNR}}$. $\qquad\square$

*Proof of Theorem 4.* Throughout the proof, we will omit the use $\lceil\cdot\rceil$ or $\lfloor\cdot\rfloor$ notation, for the sake of simplicity.

**Step 1.** We let $f_1(\cdot) = m_\Delta$ and $f_2(\cdot) = m_{\Delta+1}$, which are the before and after change point mean functions respectively. To be specific, we let $\mathcal{X} = B(0, r_{\mathcal{X}})$, with

$$r_{\mathcal{X}} = \max\left\{\left(\frac{8\sigma^2\log(1/\gamma)}{\kappa^2}\right)^{1/d}, 2\kappa\right\}.$$

We let

$$f_2(x) = f_1(x) + \begin{cases} \kappa - \|x\|, & x \in B(0,\kappa) \\ 0, & x \in \mathcal{X} \setminus B(0,\kappa), \end{cases} \quad \text{and} \quad f_1(x) = 0, \forall x \in \mathcal{X}.$$

We let the distribution of $X$ be uniform on $\mathcal{X}$ with density $p_X(x) = u = V_d^{-1}r_{\mathcal{X}}^{-d}$, for any $x \in \mathcal{X}$. We have that

$$\|f_2 - f_1\|_\infty = \kappa$$

and both $f_j$, $j = 1, 2$, are Lipschitz with constant upper bounded by 1.

**Step 2.** For any $n \in \mathbb{N}^*$, let $P^n$ be the restriction of a distribution $P$ to $\mathcal{F}_n$, i.e. the $\sigma$-field generated by the observations $\{(X_i, Y_i)\}_{i=1}^n$. For any $\nu \geq 1$ and $n \geq \nu$, we let

$$Z_{\nu,n} = \log\left(\frac{dP^n_{\kappa,\sigma,\nu}}{dP^n_{\kappa,\sigma,\infty}}\right) = \sum_{i=\nu+1}^n Z_i,$$

where $P_{\kappa,\sigma,\infty}$ indicates the joint distribution under which there is no change point and

$$Z_i = \log\left\{\frac{dP_{\kappa,\sigma,\nu}(X_i, Y_i)}{dP_{\kappa,\sigma,\nu}(X_i, Y_i)}\right\} = \frac{f_2(X_i) - f_1(X_i)}{\sigma^2}\left\{Y_i - \frac{f_1(X_i) + f_2(X_i)}{2}\right\}.$$

**Step 2.1.** For any $\nu \geq 1$, define the event

$$\mathcal{E}_\nu = \left\{\nu < T \leq \nu + \frac{\sigma^2}{\kappa^{2+d}}\log(1/\gamma),\ Z_{\nu,T} < \frac{3}{4}\log\left(\frac{1}{\gamma}\right)\right\}.$$

Then we have

$$\mathbb{P}_{\kappa,\sigma,\nu}(\mathcal{E}_\nu) = \int_{\mathcal{E}_\nu} \exp(Z_{\nu,T})\, dP_{\kappa,\sigma,\infty} \leq \gamma^{-3/4}\mathbb{P}_{\kappa,\sigma,\infty}(\mathcal{E}_\nu)$$

$$\leq \gamma^{-3/4}\mathbb{P}_{\kappa,\sigma,\infty}\left\{\nu < T \leq \nu + \frac{\sigma^2}{\kappa^{2+d}}\log(1/\gamma)\right\} \leq \gamma^{-3/4}\gamma = \gamma^{1/4}, \tag{S.3}$$

where the last inequality follows from the definition of $\mathcal{D}(\gamma)$.

**Step 2.2.** For any $\nu \geq 1$ and $T \in \mathcal{D}(\gamma)$, since $\{T \geq \nu\} \in \mathcal{F}_{\nu-1}$, we have that

$$\mathbb{P}_{\kappa,\sigma,\nu}\left\{\nu < T \leq \nu + \frac{\sigma^2}{\kappa^{2+d}}\log(1/\gamma),\ Z_{\nu,T} \geq \frac{3}{4}\log\left(\frac{1}{\gamma}\right)\Big| T > \nu\right\}$$

$$\leq \operatorname{ess\,sup} \mathbb{P}_{\kappa,\sigma,\nu}\left\{\max_{1 \leq t \leq \frac{\sigma^2}{\kappa^{2+d}}\log\left(\frac{1}{\gamma}\right)} Z_{\nu,\nu+t} \geq \frac{3}{4}\log\left(\frac{1}{\gamma}\right)\Big|(X_i, Y_i)_{i=1}^\nu\right\}$$

$$\leq \frac{\sigma^2}{\kappa^{2+d}}\log\left(\frac{1}{\gamma}\right)\max_{1 \leq t \leq \frac{\sigma^2}{\kappa^{2+d}}\log\left(\frac{1}{\gamma}\right)} \mathbb{E}_X\mathbb{P}_\varepsilon\left\{Z_{\nu,\nu+t} \geq \frac{3}{4}\log\left(\frac{1}{\gamma}\right)\Big|(X_i, Y_i)_{i=1}^\nu\right\}$$

$$= \frac{\sigma^2}{\kappa^{2+d}}\log\left(\frac{1}{\gamma}\right)\left\{\max_{1 \leq t \leq \frac{\sigma^2}{\kappa^{2+d}}\log\left(\frac{1}{\gamma}\right)} g(t) + \max_{1 \leq t \leq \frac{\sigma^2}{\kappa^{2+d}}\log\left(\frac{1}{\gamma}\right)} h(t)\right\} = (I) + (II),$$

where

$$g(t) = \mathbb{E}_X\mathbb{P}_\varepsilon\left\{\left[Z_{\nu,\nu+t} \geq \frac{3}{4}\log\left(\frac{1}{\gamma}\right)\right] \cap \left[\sum_{i=\nu+1}^{\nu+t} \frac{\{f_2(X_i) - f_1(X_i)\}^2}{2\sigma^2} \leq \frac{1}{4}\log\left(\frac{1}{\gamma}\right)\right]\right\}$$

and

$$h(t) = \mathbb{E}_X\mathbb{P}_\varepsilon\left\{\left[Z_{\nu,\nu+t} \geq \frac{3}{4}\log\left(\frac{1}{\gamma}\right)\right] \cap \left[\sum_{i=\nu+1}^{\nu+t} \frac{\{f_2(X_i) - f_1(X_i)\}^2}{2\sigma^2} > \frac{1}{4}\log\left(\frac{1}{\gamma}\right)\right]\right\}.$$

**Step 2.2.1.** We first deal with $(II)$. As for $h(t)$, we have that

$$h(t) \leq \mathbb{P}_X\left\{\sum_{i=\nu+1}^{\nu+t} \frac{\{f_2(X_i) - f_1(X_i)\}^2}{2\sigma^2} > \frac{1}{4}\log\left(\frac{1}{\gamma}\right)\right\}$$

$$\leq \mathbb{P}_X\left\{\sum_{i=\nu+1}^{\nu+\frac{\sigma^2}{\kappa^{2+d}}\log\left(\frac{1}{\gamma}\right)} \frac{\{f_2(X_i) - f_1(X_i)\}^2}{2\sigma^2} > \frac{1}{4}\log\left(\frac{1}{\gamma}\right)\right\}.$$

If

$$\frac{1}{2\kappa^d}\log\left(\frac{1}{\gamma}\right) \le \frac{1}{4}\log\left(\frac{1}{\gamma}\right),$$

i.e. $\kappa \ge 2^{1/d}$, then $h(t) = 0$. Otherwise,

$$h(t) \le V_d^t \kappa^{dt} u^t \le V_d \kappa^d u,$$

where the last inequality is due to the fact that $u V_d \kappa^d < 1$.

Then we have that

$$
\begin{aligned}
(II) &= \frac{\sigma^2}{\kappa^{2+d}}\log\left(\frac{1}{\gamma}\right) \max_{1 \le t \le \frac{\sigma^2}{\kappa^{2+d}}\log\left(\frac{1}{\gamma}\right)} h(t) = \frac{\sigma^2}{\kappa^{2+d}}\log\left(\frac{1}{\gamma}\right) V_d \kappa^d u \\
&= \frac{\sigma^2}{\kappa^2}\log\left(\frac{1}{\gamma}\right)\min\left\{\frac{\kappa^2}{8\sigma^2\log(1/\gamma)}, \frac{1}{2^d \kappa^d}\right\} \le 1/8. \quad\quad\text{(S.4)}
\end{aligned}
$$

**Step 2.2.2.** We then deal with $(I)$. As for $g(t)$, we have that

$$
\begin{aligned}
g(t) &= \mathbb{E}_X \mathbb{P}_\varepsilon\left\{\left[Z_{\nu,\nu+t} \ge \frac{3}{4}\log\left(\frac{1}{\gamma}\right)\right] \cap \left[\sum_{i=\nu+1}^{\nu+t}\frac{\{f_2(X_i)-f_1(X_i)\}^2}{2\sigma^2} \le \frac{1}{4}\log\left(\frac{1}{\gamma}\right)\right]\right\} \\
&\le \mathbb{E}_X \mathbb{P}_\varepsilon\left\{Z_{\nu,\nu+t} - \sum_{i=\nu+1}^{\nu+t}\frac{\{f_2(X_i)-f_1(X_i)\}^2}{2\sigma^2} \ge \frac{1}{2}\log\left(\frac{1}{\gamma}\right)\right\} \\
&\le \mathbb{E}_X\left[\exp\left\{-\frac{(1/2)^2\log^2(1/\gamma)}{2\sum_{i=\nu+1}^{\nu+t}\frac{\{f_2(X_i)-f_1(X_i)\}^2}{\sigma^2}}\right\}\right] \\
&= \int\cdots\int_{B(0,\kappa)^{\otimes t}}\exp\left\{-\frac{(1/2)^2\log^2(1/\gamma)}{2\sum_{i=1}^{t}\frac{\{f_2(x_i)-f_1(x_i)\}^2}{\sigma^2}}\right\}u^t\,dx_1\cdots dx_t \\
&= \int\cdots\int_{B(0,\kappa)^{\otimes t}}\exp\left\{-\frac{(1/2)^2\log^2(1/\gamma)}{2\sum_{i=1}^{t}\frac{(\kappa-\|x_i\|)^2}{\sigma^2}}\right\}u^t\,dx_1\cdots dx_t \\
&\le u^t\int\cdots\int_{B(0,\kappa)^{\otimes t}}\exp\left\{-\frac{(1/2)^2\log^2(1/\gamma)}{2t\frac{\kappa^2}{\sigma^2}}\right\}dx_1\cdots dx_t \\
&= \exp\left\{-\frac{(1/2)^2\log^2(1/\gamma)}{2t\frac{\kappa^2}{\sigma^2}}\right\}\left(uV_d\kappa^d\right)^t.
\end{aligned}
$$

Therefore

$$
\begin{aligned}
(I) &= \frac{\sigma^2}{\kappa^{2+d}}\log\left(\frac{1}{\gamma}\right)\max_{1 \le t \le \frac{\sigma^2}{\kappa^{2+d}}\log\left(\frac{1}{\gamma}\right)}g(t) \\
&\le \frac{\sigma^2}{\kappa^{2+d}}\log\left(\frac{1}{\gamma}\right)\exp\left\{-\frac{(1/2)^2\log^2(1/\gamma)}{2\frac{\sigma^2}{\kappa^{2+d}}\log\left(\frac{1}{\gamma}\right)\frac{\kappa^2}{\sigma^2}}\right\}uV_d\kappa^d \\
&\le (1/8)\exp\left\{-\frac{\kappa^d\log(1/\gamma)}{8}\right\} = (1/8)\gamma^{\kappa^d} < 1/8,
\end{aligned}
$$

where the second inequality is due to (S.4).

**Step 2.2.2.** Combining the previous two steps, we have that

$$\mathbb{P}_{\kappa,\sigma,\nu}\left\{\nu < T \le \nu + \frac{\sigma^2}{\kappa^{2+d}}\log(1/\gamma), Z_{\nu,T} \ge \frac{3}{4}\log\left(\frac{1}{\gamma}\right)\Big| T > \nu\right\} < 1/4. \quad\quad\text{(S.5)}$$

**Step 3.** Combining (S.3) and (S.5), we have that

$$\mathbb{P}_{\kappa,\sigma,\nu}\left\{\nu < T \leq \nu + \frac{\sigma^2}{\kappa^{2+d}}\log(1/\gamma)\right\} \leq \gamma^{1/4} + 1/4.$$

Since the upper bound in the above display is independent of $\nu$, we have that

$$\sup_{\nu \geq 1}\mathbb{P}_{\kappa,\sigma,\nu}\left\{\nu < T \leq \nu + \frac{\sigma^2}{\kappa^{2+d}}\log(1/\gamma)\right\} \leq \gamma^{1/4} + 1/4.$$

Therefore, for any change point time $\Delta$, we have that

$$\mathbb{E}_{\kappa,\sigma,\Delta}\{(T-\Delta)_+\} \geq \frac{\sigma^2}{\kappa^{2+d}}\log(1/\gamma)\mathbb{P}_{\kappa,\sigma,\nu}\left\{T - \nu > \frac{\sigma^2}{\kappa^{2+d}}\log(1/\gamma)\right\}$$

$$= \frac{\sigma^2}{\kappa^{2+d}}\log(1/\gamma)\left[\mathbb{P}_{\kappa,\sigma,\nu}\{T > \nu\} - \mathbb{P}_{\kappa,\sigma,\nu}\left\{\nu < T \leq \nu + \frac{\sigma^2}{\kappa^{2+d}}\log(1/\gamma)\right\}\right]$$

$$\geq \frac{\sigma^2}{\kappa^{2+d}}\log(1/\gamma)(1 - \gamma - \gamma^{1/4} - 1/4) \geq \frac{\sigma^2}{2\kappa^{2+d}}\log(1/\gamma),$$

where the last inequality holds when $\gamma + \gamma^{1/4} < 1/4$. □

*Proof of Theorem 1.* In the case that

$$\frac{s(t-s)}{t}c_{\min}^2 h^{2d}\alpha^2 \geq 64\log\left(\frac{72t^3}{\gamma c_{\min}h^d}\right),$$

we start by writing

$$|\widehat{D}_{s,t} - D_{s,t}| \leq \sqrt{\frac{s(t-s)}{t}}\left|\sup_{x\in\mathbb{R}^d}|\widehat{m}_{1:s}(x) - \widehat{m}_{(s+1):t}(x)| - \sup_{x\in\mathbb{R}^d}|m_{1:s}^{(h)}(x) - m_{(s+1):t}^{(h)}(x)|\right|$$

$$+ \left|\sqrt{\frac{s(t-s)}{t}}\sup_{x\in\mathbb{R}^d}|m_{1:s}^{(h)}(x) - m_{(s+1):t}^{(h)}(x)| - D_{s,t}\right|$$

$$= (I') + (III).$$

Here $(III)$ is the same term as appears in the proof of Theorem 3, and we do not need to bound it again here. The term $(I')$ roughly corresponds to the non-private $(I)$ and $(II)$, but new techniques must be applied. Instead of using Lemma S.5 we use Lemma S.7, and we also need to bound a truncation error, but otherwise the proof is very similar.

For regression functions $m$ we must bound the truncation error

$$\max_j \frac{1}{\mu(A_{h,j})}\left|\int_{A_{h,j}}\{m(x) - \mathbb{E}([Y]_{-M}^M|X = x)\}\mu(dx)\right|,$$

and this changes the overall proof by just adding some bias. We use the facts that $\sup_x |m(x)| \leq M_0$ and $Y|X = x$ is $\sigma$-subgaussian for all $x$. For a $\sigma$-subgaussian random variable $Z$ and $t \geq \sigma\sqrt{\log(4)}$ we have that (using Exercise 2.3 of [3])

$$\mathbb{E}(|Z|\mathbb{1}_{|Z|\geq t}) \leq t\inf_{k\in\mathbb{N}}\frac{\mathbb{E}(|Z|^k)}{t^k} \leq t\inf_{\lambda>0}\frac{\mathbb{E}(e^{\lambda|Z|})}{e^{\lambda t}} \leq 2t\inf_{\lambda\in\mathbb{R}}\frac{\mathbb{E}(e^{\lambda Z})}{e^{\lambda t}} \leq 2t\inf_{\lambda\in\mathbb{R}}e^{-\lambda t + \lambda^2\sigma^2/2} = 2te^{-\frac{t^2}{2\sigma^2}}.$$

When $M \geq M_0 + \sigma\sqrt{2\log(2 + \sigma/h) + \log\log(2 + \sigma/h)}$ we therefore have

$$\max_j \frac{1}{\mu(A_{h,j})}\left|\int_{A_{h,j}}\{m(x) - \mathbb{E}([Y]_{-M}^M|X = x)\}\mu(dx)\right|$$

$$\leq \max_j \frac{1}{\mu(A_{h,j})}\int_{A_{h,j}}\mathbb{E}\{(|Y| - M)\mathbb{1}_{|Y|\geq M}|X = x\}\mu(dx)$$

$$\leq \max_j \frac{1}{\mu(A_{h,j})}\int_{A_{h,j}}\mathbb{E}\{(|Y| - m(x))\mathbb{1}_{|Y-m(x)|\geq M-M_0}|X = x\}\mu(dx)$$

$$\leq 2(M - M_0) \exp\left(-\frac{(M - M_0)^2}{2\sigma^2}\right)$$

$$\leq 2\frac{\sigma\sqrt{2\log(2 + \sigma/h) + \log\log(2 + \sigma/h)}}{(2 + \sigma/h)\sqrt{\log(2 + \sigma/h)}} \leq 2\sqrt{2}h,$$

which is the same order as the bias. We see that we have

$$\mathbb{P}\left(|\widehat{D}_{s,t} - D_{s,t}| \geq b_{s,t}\right) \leq \gamma/(2t^3). \tag{S.6}$$

On the other hand, in the case that

$$\frac{s(t - s)}{t} c_{\min}^2 h^{2d}\alpha^2 < 64\log\left(\frac{72t^3}{\gamma c_{\min}h^d}\right),$$

we will never flag a change, and there is nothing to prove.

In the setting of Assumption 2 we see from (S.6) and a union bound argument that we have

$$\mathbb{P}\left(\max_{t\in\mathbb{N}^*}\max_{1\leq s\leq t}(\widehat{D}_{s,t} - b_{s,t}) \geq 0\right) \leq \sum_{t=1}^{\infty} t\frac{\gamma}{2t^3} = \gamma\frac{\pi^2}{12} \leq \gamma.$$

Similarly, in the setting of Assumption 3, we have

$$\mathbb{P}\left(\max_{1\leq t\leq\Delta}\max_{1\leq s\leq t}(\widehat{D}_{s,t} - b_{s,t}) \geq 0\right) \leq \sum_{t=1}^{\Delta} t\frac{\gamma}{2t^3} \leq \gamma.$$

We have that

$$\frac{\Delta\epsilon}{\Delta + \epsilon} c_{\min}^2 h^{2d}\alpha^2 \geq 64\log\left(\frac{72t^3}{\gamma c_{\min}h^d}\right)$$

when $C_\epsilon$ and $C_{\text{SNR}}$ are chosen large enough. Thus, by (S.6) and Lemma S.1, with probability at least $1 - \gamma$, we have

$$\widehat{D}_{\Delta,\Delta+\epsilon} - b_{\Delta,\Delta+\epsilon} \geq D_{\Delta,\Delta+\epsilon} - 2b_{\Delta,\Delta+\epsilon}$$

$$\geq \kappa\sqrt{\frac{\Delta\epsilon}{\Delta + \epsilon}} - 4\sqrt{\frac{\Delta\epsilon}{\Delta + \epsilon}}(C_{\text{Lip}}\sqrt{d} + 2\sqrt{2})h - \frac{2M}{c_{\min}h^d\alpha}\sqrt{\log\left(\frac{72t^3}{\gamma c_{\min}h^d}\right)} > 0,$$

where the final inequality holds when $\kappa/h$ is above a constant threshold and $C_{\text{SNR}}$ is large enough.

$\square$

*Proof of Theorem 2.* Throughout the proof, we will omit the use $\lceil\cdot\rceil$ or $\lfloor\cdot\rfloor$ notation, for the sake of simplicity.

**Step 1.** Let $Q$ be any sequentially interactive privacy mechanism, with output in some space $\mathcal{W}$, whose density satisfies

$$\frac{q_i(w|x, y, w_{i-1}, \ldots, w_1)}{q_i(w|x', y', w_{i-1}, \ldots, w_1)} \leq e^\alpha, \quad \forall w, w_1, \ldots, w_{i-1} \in \mathcal{W}, x, x' \in \mathbb{R}^d, y, y' \in \mathbb{R}.$$

Let $f_1(\cdot) = m_\Delta$ and $f_2(\cdot) = m_{\Delta+1}$, which are the before and after change point mean functions respectively. Let $P_X(\cdot)$ be the distribution of $X$, staying the same before and after the change point. Let $Y_i = m_i(X) + \text{Unif}[-\sigma, \sigma]$, for any $i \in \mathbb{N}^*$. Let $P_{k|X}$, $k = 1, 2$, denote the distribution of $Y$ given $X$, before and after the change point. Let $h_{i,k}(w|w_{i-1}, \ldots, w_1) = \int q(w|x, y, w_{i-1}, \ldots, w_1)\, dP_X(x) dP_{k|X}(y|x)$. Let $\mathcal{X} = B(0, r_\mathcal{X})$, with

$$r_\mathcal{X} = \left(\frac{2(e^\alpha - 1)}{\alpha}\right)^{1/d}. \tag{S.7}$$

We let

$$f_2(x) = f_1(x) + \begin{cases} \kappa - \|x\|, & x \in B(0, \kappa) \\ 0, & x \in \mathcal{X} \setminus B(0, \kappa), \end{cases} \quad \text{and} \quad f_1(x) = 0, \forall x \in \mathcal{X}.$$

We let the distribution of $X$ be uniform on $\mathcal{X}$ with density $p_X(x) = u = V_d^{-1} r_{\mathcal{X}}^{-d}$, for any $x \in \mathcal{X}$. We have that
$$\|f_2 - f_1\|_\infty = \kappa$$
and both $f_j$, $j = 1, 2$, are Lipschitz with constant upper bounded by 1.

**Step 2.** For any $n \in \mathbb{N}^*$, let $P^n$ be the restriction of a distribution $P$ to $\mathcal{F}_n$, i.e. the $\sigma$-field generated by the observations $\{W_i\}_{i=1}^n \subset \mathcal{W}$. For any $\nu \geq 1$ and $n \geq \nu$, we let

$$Z_{\nu,n} = \log\left(\frac{dP_{\kappa,\sigma,\nu}^n}{dP_{\kappa,\sigma,\infty}^n}\right) = \sum_{i=\nu+1}^{n} Z_i = \sum_{i=\nu+1}^{n} \log\left\{\frac{h_{i,2}(W_i|W_{i-1},\ldots,W_1)}{h_{i,1}(W_i|W_{i-1},\ldots,W_1)}\right\},$$

where $P_{\kappa,\sigma,\infty}$ indicates the joint distribution under which there is no change point.

It follows from Lemma 1 in [2, Supplementary material], that we have

$$|Z_i| \leq \min\{2, e^\alpha\}(e^\alpha - 1)d_{\mathrm{TV}}(P_1, P_2), \quad i \in \{\nu+1,\ldots,n\}.$$

where $P_1$ and $P_2$ are the joint distributions of $(X, Y)$ before and after the change point. Moreover, by calculations around Lemma 1 in [2, Supplementary material] we have

$$0 \leq \int h_{i,2}(w|w_{i-1},\ldots,w_1) \log\frac{h_{i,2}(w|w_{i-1},\ldots,w_1)}{h_{i,1}(w|w_{i-1},\ldots,w_1)}\, du \leq \min(4, e^{2\alpha})(e^\alpha - 1)^2 d_{\mathrm{TV}}(P_1, P_2)^2$$

for all $w_1,\ldots,w_{i-1}$. Since

$$d_{\mathrm{TV}}(P_1, P_2) = \frac{1}{2\sigma}\int_{B(0,\kappa)}(\kappa - \|x\|)p_X(x)\, dx \leq \frac{\kappa}{2\sigma}V_d\kappa^d u,$$

we have that for any $i > \nu$, $|Z_i| \leq \min\{2, e^\alpha\}(e^\alpha - 1)\frac{\kappa}{2\sigma}V_d\kappa^d u \leq \alpha\kappa^{d+1}/(2\sigma)$, and $\mathbb{E}(Z_i|W_{i-1},\ldots,W_1) \leq \alpha^2\kappa^{2d+2}/(4\sigma^2)$ almost surely.

**Step 2.1.** For any $\nu \geq 1$, define the event

$$\mathcal{E}_\nu = \left\{\nu < T \leq \nu + \frac{\sigma^2\log(1/\gamma)}{\kappa^{2+2d}\alpha^2}, \; Z_{\nu,T} < \frac{3}{4}\log\left(\frac{1}{\gamma}\right)\right\}.$$

Then we have

$$\mathbb{P}_{\kappa,\sigma,\nu}(\mathcal{E}_\nu) = \int_{\mathcal{E}_\nu} \exp\left(Z_{\nu,T}\right)dP_{\kappa,\sigma,\infty} \leq \gamma^{-3/4}\mathbb{P}_{\kappa,\sigma,\infty}(\mathcal{E}_\nu)$$

$$\leq \gamma^{-3/4}\mathbb{P}_{\kappa,\sigma,\infty}\left\{\nu < T \leq \nu + \frac{\sigma^2\log(1/\gamma)}{\kappa^{2+2d}\alpha^2}\right\} \leq \gamma^{-3/4}\gamma = \gamma^{1/4}, \qquad \text{(S.8)}$$

where the last inequality follows from the definition of $\mathcal{D}(\gamma)$.

**Step 2.2.** For any $\nu \geq 1$ and $T \in \mathcal{D}(\gamma)$, since $\{T \geq \nu\} \in \mathcal{F}_{\nu-1}$, we have that

$$\mathbb{P}_{\kappa,\sigma,\nu}\left\{\nu < T \leq \nu + \frac{\sigma^2\log(1/\gamma)}{\kappa^{2+2d}\alpha^2}, \; Z_{\nu,T} \geq \frac{3}{4}\log\left(\frac{1}{\gamma}\right)\Big|T > \nu\right\}$$

$$\leq \operatorname*{ess\,sup}\mathbb{P}_{\kappa,\sigma,\nu}\left\{\max_{1 \leq t \leq \frac{\sigma^2\log(1/\gamma)}{\kappa^{2+2d}\alpha^2}} Z_{\nu,\nu+t} \geq \frac{3}{4}\log\left(\frac{1}{\gamma}\right)\Big|(W_i)_{i=1}^\nu\right\}$$

$$\leq \frac{\sigma^2\log(1/\gamma)}{\kappa^{2+2d}\alpha^2}\max_{1 \leq t \leq \frac{\sigma^2\log(1/\gamma)}{\kappa^{2+2d}\alpha^2}}\mathbb{P}_{\kappa,\sigma,\nu}\left\{Z_{\nu,\nu+t} \geq \frac{3}{4}\log\left(\frac{1}{\gamma}\right)\Big|(W_i)_{i=1}^\nu\right\}.$$

$$= \frac{\sigma^2\log(1/\gamma)}{\kappa^{2+2d}\alpha^2}\max_{1 \leq t \leq \frac{\sigma^2\log(1/\gamma)}{\kappa^{2+2d}\alpha^2}} g(t),$$

where

$$g(t) = \mathbb{P}_{\kappa,\sigma,\nu}\left\{Z_{\nu,\nu+t} \geq \frac{3}{4}\log\left(\frac{1}{\gamma}\right)\right\}.$$

As for $g(t)$, we have that

$$g(t) \leq \mathbb{P}_{\kappa,\sigma,\nu} \left\{ \sum_{i=\nu+1}^{\nu+t} \{Z_i - \mathbb{E}(Z_i|W_{i-1},\ldots,W_1)\} \geq \frac{3}{4} \log\left(\frac{1}{\gamma}\right) - \frac{\sigma^2 \log(1/\gamma)}{\kappa^{2+2d}\alpha^2} \frac{\alpha^2\kappa^{2d+2}}{4\sigma^2} \right\}$$

$$\leq \mathbb{P}_{\kappa,\sigma,\nu} \left\{ \sum_{i=\nu+1}^{\nu+t} \{Z_i - \mathbb{E}(Z_i|W_{i-1},\ldots,W_1)\} \geq \frac{1}{2} \log\left(\frac{1}{\gamma}\right) \right\}$$

$$\leq \exp\left\{ -\frac{\log^2(1/\gamma)}{\frac{\sigma^2\log(1/\gamma)}{\kappa^{2+2d}\alpha^2}\alpha^2\kappa^{2d+2}/\sigma^2} \right\} = \gamma,$$

where the second inequality is due the Azuma–Hoeffding inequality [e.g. 3, Corollary 2.20]. Therefore

$$\mathbb{P}_{\kappa,\sigma,\nu} \left\{ \nu < T \leq \nu + \frac{\sigma^2}{\kappa^{2+d}} \log(1/\gamma),\, Z_{\nu,T} \geq \frac{3}{4} \log\left(\frac{1}{\gamma}\right) \Big| T > \nu \right\} \leq \frac{\sigma^2\log(1/\gamma)}{\kappa^{2+2d}\alpha^2} \gamma \leq \gamma^{1/4}.$$

**Step 3.** Combining the above we have that

$$\mathbb{P}_{\kappa,\sigma,\nu} \left\{ \nu < T \leq \nu + \frac{\sigma^2}{\kappa^{2+d}} \log(1/\gamma),\, Z_{\nu,T} \geq \frac{3}{4} \log\left(\frac{1}{\gamma}\right) \Big| T > \nu \right\} < 2\gamma^{1/4}.$$

Since the upper bound in the above display is independent of $\nu$, we have that

$$\sup_{\nu \geq 1} \mathbb{P}_{\kappa,\sigma,\nu} \left\{ \nu < T \leq \nu + \frac{\sigma^2}{\kappa^{2+d}} \log(1/\gamma) \right\} \leq 2\gamma^{1/4}.$$

Therefore, for any change point time $\Delta$, we have that

$$\mathbb{E}_{\kappa,\sigma,\Delta}\{(T-\Delta)_+\} \geq \frac{\sigma^2\log(1/\gamma)}{\kappa^{2+2d}\alpha^2} \mathbb{P}_{\kappa,\sigma,\nu}\left\{ T - \nu > \frac{\sigma^2\log(1/\gamma)}{\kappa^{2+2d}\alpha^2} \right\}$$

$$= \frac{\sigma^2\log(1/\gamma)}{\kappa^{2+2d}\alpha^2} \left[ \mathbb{P}_{\kappa,\sigma,\nu}\{T > \nu\} - \mathbb{P}_{\kappa,\sigma,\nu}\left\{ \nu < T \leq \nu + \frac{\sigma^2\log(1/\gamma)}{\kappa^{2+2d}\alpha^2} \right\} \right]$$

$$\geq \frac{\sigma^2\log(1/\gamma)}{\kappa^{2+2d}\alpha^2}(1 - \gamma - 2\gamma^{1/4}) \geq \frac{\sigma^2\log(1/\gamma)}{2\kappa^{2+2d}\alpha^2}.$$

$\square$

## S.2  Auxiliary lemmas

### S.2.1  Population quantities

**Lemma S.1.** *For $D_{s,t}$ defined in* (S.1)*, it holds that*

$$D_{s,t} = \begin{cases} 0, & 1 \leq s < t \leq \Delta, \\ \kappa\sqrt{\Delta}\sqrt{\frac{t-\Delta}{t}}, & s = \Delta < t. \end{cases}$$

*Proof.* When $1 \leq s < t \leq \Delta$, by definition, we have that $D_{s,t} = 0$.

When $s = \Delta < t$, let $m_\Delta = f_1$ and $m_{\Delta+1} = f_2$. Note that

$$D_{\Delta,t} = \sqrt{\frac{\Delta(t-\Delta)}{t}} \|f_1 - f_2\|_\infty = \kappa\sqrt{\Delta}\sqrt{\frac{t-\Delta}{t}},$$

where the last identity follows from Assumption 3. $\square$

**Lemma S.2.** *When $1 \leq s < t \leq \Delta$, it holds that*

$$\mathbb{P}\left\{ \left| \max_{i=1}^{t} D_{s,t}^{(h)}(X_i) - D_{s,t}^{(h)} \right| = 0 \right\} = 1.$$

*When $s = \Delta < t$, it holds that*

$$\mathbb{P}\left\{ \left| \max_{i=1}^{t} D_{s,t}^{(h)}(X_i) - D_{s,t}^{(h)} \right| > 0 \right\} \leq \frac{1}{c_{\min}h^d} \exp(-\Delta c_{\min}h^d)$$

*Proof.* **Case 1.** When $1 \leq s < t \leq \Delta$, by definition, we have that

$$D_{s,t}^{(h)}(x) = 0, \quad x \in \mathbb{R}^d.$$

Then we have that

$$\mathbb{P} \left\{ \left| \max_{i=1}^{t} D_{s,t}^{(h)}(X_i) - D_{s,t}^{(h)} \right| = 0 \right\} = 1.$$

**Case 2.** The only way that we can have $\max_{i=1}^{t} D_{s,t}^{(h)}(X_i) \neq D_{s,t}^{(h)}$ is if there exists some $A_{h,j}$ with no observations falling within it. We have that

$$1 = \sum_j \mu(A_{h,j}) \geq c_{\min} h^d N_h$$

Thus, when $s = \Delta < t$, using a union bound, it holds that

$$\mathbb{P} \left\{ \left| \max_{i=1}^{t} D_{s,t}^{(h)}(X_i) - D_{s,t}^{(h)} \right| > 0 \right\} \leq \mathbb{P} \left\{ \sum_{i=1}^{t} \mathbb{1}_{\{X_i \in A_{h,j}\}} = 0 \text{ for some } j \right\}$$

$$\leq \frac{1}{c_{\min} h^d} \max_j \{1 - \mu(A_{h,j})\}^t$$

$$\leq \frac{1}{c_{\min} h^d} \exp(-\Delta c_{\min} h^d),$$

as required.

$\square$

**Lemma S.3.** *For any integer pairs $(s,t)$, $1 \leq s < t \leq \Delta$, it holds that $|D_{s,t}^{(h)} - D_{s,t}| = 0$. When $s = \Delta < t$, it holds that*

$$|D_{s,t}^{(h)} - D_{s,t}| \leq 2\sqrt{\frac{s(t-s)}{t}} C_{\mathrm{Lip}} \sqrt{d} h.$$

*Proof.* For any integer pairs $(s,t)$, $1 \leq s < t \leq \Delta$, it holds that

$$m_{1:s}^{(h)}(x) = m_{(s+1):t}^{(h)}(x) \quad \text{and} \quad m_{1:s}(x) = m_{(s+1):t}(x), \quad \forall x \in \mathbb{R}^d,$$

which implies that $|D_{s,t}^{(h)} - D_{s,t}| = 0$.

When $s = \Delta < t$, it holds that

$$|D_{\Delta,t}^{(h)} - D_{\Delta,t}|$$

$$= \sqrt{\frac{\Delta(t-\Delta)}{t}} \left| \sup_{x \in \mathbb{R}^p} |m_{\Delta}^{(h)}(x) - m_{\Delta+1}^{(h)}(x)| - \sup_{x \in \mathbb{R}^p} |m_{\Delta}(x) - m_{\Delta+1}(x)| \right|$$

$$\leq \sqrt{\frac{\Delta(t-\Delta)}{t}} \sup_{x \in \mathbb{R}^p} \left| |m_{\Delta}^{(h)}(x) - m_{\Delta+1}^{(h)}(x)| - |m_{\Delta}(x) - m_{\Delta+1}(x)| \right|$$

$$\leq 2\sqrt{\frac{\Delta(t-\Delta)}{t}} \max \left\{ \sup_{x \in \mathbb{R}^p} |m_{\Delta}^{(h)}(x) - m_{\Delta}(x)|, \sup_{x \in \mathbb{R}^p} |m_{\Delta+1}^{(h)}(x) - m_{\Delta+1}(x)| \right\}$$

$$\leq 2\sqrt{\frac{\Delta(t-\Delta)}{t}} C_{\mathrm{Lip}} \sqrt{d} h,$$

where the last inequality follows Assumption 1.

$\square$

### S.2.2 Sample terms

**Lemma S.4.** *Let $n \in \mathbb{N}, p \in (0,1)$ and $x \in (0,1]$. Let $(B_1, \ldots, B_n)$ be an independent sequence of $\mathrm{Ber}(p)$ random variables, and let $(\epsilon_1, \ldots, \epsilon_n)$ be a sequence of real-valued random variables such that, conditionally on $(B_1, \ldots, B_n)$, it holds*

(i) $\epsilon_1, \ldots, \epsilon_n$ are independent; and

(ii) $\mathbb{E}(e^{\lambda \epsilon_i}) \le e^{\lambda^2/2}$ for all $\lambda \in \mathbb{R}$.

*Then we have*

$$\mathbb{P}\left( \left| \sum_{i=1}^n \epsilon_i B_i \right| \ge x \sum_{i=1}^n B_i \right) \le 2e^{-npx^2/4}.$$

*Proof.* Writing $N = \sum_{i=1}^n B_i$, we may condition on $(B_1, \ldots, B_n)$ to see that

$$\mathbb{P}\left( \frac{1}{N} \left| \sum_{i=1}^n \epsilon_i B_i \right| \ge x \right) \le 2\mathbb{E}\left[ \mathbb{1}_{\{N \ge 1\}} e^{-\frac{Nx^2}{2}} \right]$$

$$= 2\left\{ \left( 1 - p + pe^{-\frac{x^2}{2}} \right)^n - (1-p)^n \right\}$$

$$= 2\left\{ \left( 1 - p(1 - e^{-\frac{x^2}{2}}) \right)^n - (1-p)^n \right\}.$$

For $x \in (0, 1]$ we have $1 - e^{-x^2/2} \ge x^2/4$, so that

$$\mathbb{P}\left( \frac{1}{N} \left| \sum_{i=1}^n \epsilon_i B_i \right| \ge x \right) \le 2\left\{ \left( 1 - \frac{px^2}{4} \right)^n - (1-p)^n \right\} \le 2e^{-npx^2/4}.$$

$\square$

**Lemma S.5.** *Under Assumption 1, for any integer pair $(s, t)$, $1 \le s < t$, we let*

$$W_{s,t} = \sqrt{\frac{s(t-s)}{t}} \left| \max_{i=1}^t |\widetilde{m}_{1:s}(X_i) - \widetilde{m}_{(s+1):t}(X_i)| - \max_{i=1}^t |m_{1:s}^{(h)}(X_i) - m_{(s+1):t}^{(h)}(X_i)| \right|.$$

*Define the following three scenarios:*

(i) *There exists an integer pair $(s, t)$, $1 \le s < t$, such that*

$$W_{s,t} > 2\sqrt{\frac{s(t-s)}{t}} C_{\text{Lip}} \sqrt{d}h + \frac{4\sigma}{\sqrt{c_{\min}h^d}} \sqrt{5\log(t) + \log(16/\gamma)}. \tag{S.9}$$

(ii) *There exists an integer pair $(s, t)$, $1 \le s < t \le \Delta$, such that (S.9) holds.*

(iii) *Under Assumption 3, there exists an integer $t$, $t > \Delta$, such that*

$$W_{\Delta,t} > 2\sqrt{\frac{\Delta(t-\Delta)}{t}} C_{\text{Lip}} \sqrt{d}h + \frac{4\sigma}{\sqrt{c_{\min}h^d}} \sqrt{5\log(t) + \log(16/\gamma)}.$$

*We have that*

- *under Assumption 2, (i) holds with probability at most $\gamma$; and*

- *under Assumption 3, (ii) and (iii) hold with probability at most $\gamma$.*

*Proof.* For any integer pairs $1 \le s < t$ and any $\eta, \eta', \eta'' > 0$, $\eta' + \eta'' = \eta$, it holds that

$$\mathbb{P}\left( \sqrt{\frac{s(t-s)}{t}} \left| \max_{i=1}^t |\widetilde{m}_{1:s}(X_i) - \widetilde{m}_{(s+1):t}(X_i)| - \max_{i=1}^t |m_{1:s}^{(h)}(X_i) - m_{(s+1):t}^{(h)}(X_i)| \right| \ge \eta \right)$$

$$\le \mathbb{P}\left( \sqrt{\frac{s(t-s)}{t}} \max_{i=1}^t \left| |\widetilde{m}_{1:s}(X_i) - \widetilde{m}_{(s+1):t}(X_i)| - |m_{1:s}^{(h)}(X_i) - m_{(s+1):t}^{(h)}(X_i)| \right| \ge \eta \right)$$

$$\le \mathbb{P}\left( \sqrt{\frac{s(t-s)}{t}} \left\{ \max_{i=1}^t |\widetilde{m}_{1:s}(X_i) - m_{1:s}^{(h)}(X_i)| + \max_{i=1}^t |\widetilde{m}_{(s+1):t}(X_i) - m_{(s+1):t}^{(h)}(X_i)| \right\} \ge \eta \right)$$

$$\leq \mathbb{P}\left(\sqrt{\frac{s(t-s)}{t}}\max_{i=1}^{t}|\widetilde{m}_{1:s}(X_i) - m_{1:s}^{(h)}(X_i)| \geq \eta'\right)$$

$$+ \mathbb{P}\left(\sqrt{\frac{s(t-s)}{t}}\max_{i=1}^{t}|\widetilde{m}_{(s+1):t}(X_i) - m_{(s+1):t}^{(h)}(X_i)| \geq \eta''\right)$$

$$= (I) + (II). \tag{S.10}$$

**Case 1.** In this case, we consider $t \leq \Delta$. We first focus on term $(I)$.

When $x \in A_{h,j}$ with a certain $j \in \mathbb{N}$, we have

$$m_{1:s}^{(h)}(x) = \frac{\sum_{i=1}^{s}\mathbb{E}(Y_i\mathbb{1}_{\{X_i \in A_{h,j}\}})}{s\mu(A_{h,j})} = \frac{\int_{A_{h,j}} m_\Delta(x)\mu(dx)}{\mu(A_{h,j})}.$$

Then, with the convention that $0/0 = 0$, we have

$$\max_{i=1}^{t}|\widetilde{m}_{1:s}(X_i) - m_{1:s}^{(h)}(X_i)| = \max_j\left|\frac{\nu_{1:s}(A_{h,j})}{\mu_{1:s}(A_{h,j})} - \frac{\int_{A_{h,j}} m_\Delta(x)\mu(dx)}{\mu(A_{h,j})}\right|$$

$$= \max_j\left|\frac{1}{\mu_{1:s}(A_{h,j})}\sum_{i=1}^{s}\{Y_i - m_i(X_i)\}\mathbb{1}_{\{X_i \in A_{h,j}\}}\right.$$

$$\left. + \frac{1}{\mu_{1:s}(A_{h,j})}\sum_{i=1}^{s}\left(m_i(X_i) - \frac{\int_{A_{h,j}} m_\Delta(x)\mu(dx)}{\mu(A_{h,j})}\right)\mathbb{1}_{\{X_i \in A_{h,j}\}}\right|$$

$$\leq \max_j\frac{1}{\mu_{1:s}(A_{h,j})}\left|\sum_{i=1}^{s}\{Y_i - m_i(X_i)\}\mathbb{1}_{\{X_i \in A_{h,j}\}}\right|$$

$$+ \max_j\frac{1}{\mu_{1:s}(A_{h,j})}\left|\sum_{i=1}^{s}\left(m_i(X_i) - \frac{\int_{A_{h,j}} m_\Delta(x)\mu(dx)}{\mu(A_{h,j})}\right)\mathbb{1}_{\{X_i \in A_{h,j}\}}\right|$$

$$= (I.1) + (I.2). \tag{S.11}$$

As for the term $(I.1)$, we apply Lemma S.4 and take $B_i = \mathbb{1}_{\{X_i \in A_{h,j}\}}$. For any $\eta_1 \in (0, \sigma]$ we have

$$\mathbb{P}\left(\frac{1}{\mu_{1:s}(A_{h,j})}\left|\sum_{i=1}^{s}\{Y_i - m_i(X_i)\}\mathbb{1}_{\{X_i \in A_{h,j}\}}\right| \geq \eta_1\right) \leq 2\exp\left(-\frac{s\mu(A_{h,j})\eta_1^2}{4\sigma^2}\right). \tag{S.12}$$

As for the term $(I.2)$, when $x \in A_{h,j}$, due to Assumption 1, we have

$$\left|m_i(x) - \frac{\int_{A_{h,j}} m_i(x')\mu(dx')}{\mu(A_{h,j})}\right| \leq C_{\text{Lip}}\text{diam}(A_{h,j}) = C_{\text{Lip}}\sqrt{d}h,$$

hence with probability one, it holds that

$$\frac{1}{\mu_{1:s}(A_{h,j})}\left|\sum_{i=1}^{s}\left(m_i(X_i) - \frac{\int_{A_{h,j}} m_\Delta(x)\mu(dx)}{\mu(A_{h,j})}\right)\mathbb{1}_{\{X_i \in A_{h,j}\}}\right|$$

$$\leq \frac{1}{\mu_{1:s}(A_{h,j})}\sum_{i=1}^{s}\left|m_i(X_i) - \frac{\int_{A_{h,j}} m_\Delta(x)\mu(dx)}{\mu(A_{h,j})}\right|\mathbb{1}_{\{X_i \in A_{h,j}\}} \leq C_{\text{Lip}}\sqrt{d}h. \tag{S.13}$$

Combining (S.11), (S.12) and (S.13), with a union bound argument, we have that, for any $\eta_1 \in (0, \sigma]$

$$\mathbb{P}\left\{\sqrt{\frac{s(t-s)}{t}}\max_{i=1}^{t}|\widetilde{m}_{1:s}(X_i) - m_{1:s}^{(h)}(X_i)| \geq \sqrt{\frac{s(t-s)}{t}}\left(\eta_1 + C_{\text{Lip}}\sqrt{d}h\right)\right\}$$

$$\leq 2t\exp\left(-\frac{s\mu(A_{h,j})\eta_1^2}{4\sigma^2}\right). \tag{S.14}$$

For any $\eta_2 \in (0, \sigma]$, almost identical arguments lead to

$$\mathbb{P}\left\{\sqrt{\frac{s(t-s)}{t}} \max_{i=1}^{t} |\widetilde{m}_{(s+1):t}(X_i) - m_{(s+1):t}^{(h)}(X_i)| \geq \sqrt{\frac{s(t-s)}{t}} \left(\eta_2 + C_{\text{Lip}}\sqrt{d}h\right)\right\}$$

$$\leq 2t \exp\left(-\frac{(t-s)\mu(A_{h,j})\eta_2^2}{4\sigma^2}\right). \tag{S.15}$$

Combining (S.10), (S.14) and (S.15), we have that

$$\mathbb{P}\left\{\sqrt{\frac{s(t-s)}{t}} \left|\max_{i=1}^{t} |\widetilde{m}_{1:s}(X_i) - \widetilde{m}_{(s+1):t}(X_i)| - \max_{i=1}^{t} |m_{1:s}^{(h)}(X_i) - m_{(s+1):t}^{(h)}(X_i)|\right|\right.$$

$$\left. \geq \sqrt{\frac{s(t-s)}{t}}(\eta_1 + \eta_2) + 2\sqrt{\frac{s(t-s)}{t}}C_{\text{Lip}}\sqrt{d}h\right\}$$

$$\leq 2t \exp\left(-\frac{s\mu(A_{h,j})\eta_1^2}{4\sigma^2}\right) + 2t \exp\left(-\frac{(t-s)\mu(A_{h,j})\eta_2^2}{4\sigma^2}\right)$$

$$\leq 2t \exp\left(-\frac{sc_{\min}h^d\eta_1^2}{4\sigma^2}\right) + 2t \exp\left(-\frac{(t-s)c_{\min}h^d\eta_2^2}{4\sigma^2}\right),$$

where the last inequality follows from Assumption 1.

We let

$$Q_{s,t} = \sqrt{\frac{s(t-s)}{t}} \left|\max_{i=1}^{t} |\widetilde{m}_{1:s}(X_i) - \widetilde{m}_{(s+1):t}(X_i)| - \max_{i=1}^{t} |m_{1:s}^{(h)}(X_i) - m_{(s+1):t}^{(h)}(X_i)|\right|$$

$$- 2\sqrt{\frac{s(t-s)}{t}}C_{\text{Lip}}\sqrt{d}h,$$

$$\eta_1 = \sqrt{\frac{4\sigma^2}{sc_{\min}h^d}}\varepsilon_t \quad \text{and} \quad \eta_2 = \sqrt{\frac{4\sigma^2}{(t-s)c_{\min}h^d}}\varepsilon_t,$$

with $\varepsilon_t > 0$ to be specified. Therefore

$$\mathbb{P}\left\{Q_{s,t} \geq \frac{4\sigma\varepsilon_t}{\sqrt{c_{\min}h^d}}\right\} \leq \mathbb{P}\left\{Q_{s,t} \geq \sqrt{\frac{s(t-s)}{t}}(\eta_1 + \eta_2)\right\} \leq 4t \exp(-\varepsilon_t^2).$$

**Case 1.1** When $\Delta = \infty$, we have that

$$\mathbb{P}\left\{\exists s, t \in \mathbb{N}^*, t > 1, s \in [1, t) : Q_{s,t} \geq \frac{4\sigma\varepsilon_t}{\sqrt{c_{\min}h^d}}\right\}$$

$$\leq \sum_{j=1}^{\infty} \mathbb{P}\left\{\max_{2^j \leq t < 2^{j+1}} \max_{1 \leq s < t} Q_{s,t} \geq \frac{4\sigma\varepsilon_t}{\sqrt{c_{\min}h^d}}\right\} \leq \sum_{j=1}^{\infty} 2^j \max_{2^j \leq t < 2^{j+1}} \mathbb{P}\left\{\max_{1 \leq s < t} Q_{s,t} \geq \frac{4\sigma\varepsilon_t}{\sqrt{c_{\min}h^d}}\right\}$$

$$\leq \sum_{j=1}^{\infty} 2^j \max_{2^j \leq t < 2^{j+1}} 4t \max_{1 \leq s < t} \mathbb{P}\left\{Q_{s,t} \geq \frac{4\sigma\varepsilon_t}{\sqrt{c_{\min}h^d}}\right\} \leq 4\sum_{j=1}^{\infty} 2^j \max_{2^j \leq t < 2^{j+1}} t^2 \exp(-\varepsilon_t^2)$$

$$\leq 4\sum_{j=1}^{\infty} 2^{3j+2} \exp(-\varepsilon_{2^j}^2) \leq \gamma,$$

where the last inequality holds by taking

$$\varepsilon_t = \sqrt{5\log(t) + \log(16/\gamma)}. \tag{S.16}$$

**Case 1.2** When $\Delta < \infty$, with $\varepsilon_t$ defined in (S.16), it holds that

$$\mathbb{P}_\Delta\left\{\exists s, t \in \mathbb{N}^*, 1 \leq s < t \leq \Delta : Q_{s,t} \geq \frac{4\sigma\varepsilon_t}{\sqrt{c_{\min}h^d}}\right\}$$

$$\leq \mathbb{P}_\infty \left\{ \exists s, t \in \mathbb{N}^*, t > 1, s \in [1, t) : Q_{s,t} \geq \frac{4\sigma\varepsilon_t}{\sqrt{c_{\min}h^d}} \right\} \leq \gamma.$$

**Case 2.** In this case, we consider $s = \Delta < t$. Note that for any such integer pair $(s, t)$, within both intervals $[1 : s]$ and $[(s+1) : t]$, there is one and only one underlying distribution. Therefore, based on identical arguments as those in **Case 1.**, we have that

$$\mathbb{P} \left\{ Q_{\Delta,t} \geq \frac{4\sigma\varepsilon_t}{\sqrt{c_{\min}h^d}} \right\} \leq 4t \exp(-\varepsilon_t^2).$$

Then we have

$$\mathbb{P}_\Delta \left\{ \exists t \in \mathbb{N}^*, t > \Delta : Q_{\Delta,t} \geq \frac{4\sigma\varepsilon_t}{\sqrt{c_{\min}h^d}} \right\} \leq \gamma.$$

We then complete the proof. $\qquad\square$

**Lemma S.6** (Laplace concentration). *Let $\epsilon_1, \ldots, \epsilon_n$ be independent standard Laplace-distributed random variables (mean zero, variance 2). Then for all $x > 0$ we have*

$$\mathbb{P} \left( \frac{1}{n} \sum_{i=1}^n \epsilon_i \geq x \right) \leq \exp\left( -\frac{3nx^2}{4 + 3x} \right).$$

Note that this implies Lemma 1 of [1].

*Proof.* Using a Chernoff bound and taking $\lambda = \frac{n}{x}(\sqrt{1+x^2} - 1)$ we have

$$\mathbb{P} \left( \frac{1}{n} \sum_{i=1}^n \epsilon_i \geq x \right) \leq \inf_{\lambda \in (0,n)} e^{-\lambda x} \left( 1 - \frac{\lambda^2}{n^2} \right)^{-n}$$

$$\leq \exp\left( -n(\sqrt{1+x^2} - 1) - n\log\left( 1 - \frac{2 + x^2 - 2\sqrt{1+x^2}}{x^2} \right) \right)$$

$$= \exp\left( -n \left\{ \frac{x^2}{1 + \sqrt{1+x^2}} - \log\left( \frac{2}{1 + \sqrt{1+x^2}} \right) \right\} \right).$$

It is a fact (checked numerically) that

$$\frac{x^2}{1 + \sqrt{1+x^2}} - \log\left( \frac{2}{1 + \sqrt{1+x^2}} \right) \geq \frac{3x^2}{4 + 3x}$$

for all $x \geq 0$, and the result follows. $\qquad\square$

**Lemma S.7** (Private version of Lemma S.5). *Suppose that*

$$\frac{s(t-s)}{t} c_{\min}^2 h^{2d}\alpha^2 \geq 64\log\left( \frac{72t^3}{\gamma c_{\min}h^d} \right).$$

*Then, if either $1 \leq s < t \leq \Delta$ or $\Delta = s < t$, with probability at least $1 - \gamma/(2t^3)$ we have*

$$\sqrt{\frac{s(t-s)}{t}} \left| \sup_{x \in \mathbb{R}^d} |\widehat{m}_{1:s}(x) - \widehat{m}_{(s+1):t}(x)| - \sup_{x \in \mathbb{R}^d} |m_{1:s}^{(h)}(x) - m_{(s+1):t}^{(h)}(x)| \right|$$

$$\leq 2\sqrt{\frac{s(t-s)}{t}} C_{\text{Lip}}\sqrt{d}h + \frac{2^9 M}{c_{\min}h^d\alpha} \sqrt{\log\left( \frac{72t^3}{\gamma c_{\min}h^d} \right)}.$$

*Proof.* We start by writing

$$\left| \sup_{x \in \mathbb{R}^d} |\widehat{m}_{1:s}(x) - \widehat{m}_{(s+1):t}(x)| - \sup_{x \in \mathbb{R}^d} |m_{1:s}^{(h)}(x) - m_{(s+1):t}^{(h)}(x)| \right|$$

$$\leq \sup_{x\in\mathbb{R}^d}|\widehat{m}_{1:s}(x)-m_{1:s}^{(h)}(x)| + \sup_{x\in\mathbb{R}^d}|\widehat{m}_{(s+1):t}(x)-m_{(s+1):t}^{(h)}(x)|$$

As in the proof of Lemma S.5, it suffices to consider the first term in the case that $s\leq\Delta$. This is given by

$$\max_j\left|\frac{\widehat{\nu}_{1:s}(A_{h,j})}{\widehat{\mu}_{1:s}(A_{h,j})}\mathbb{1}_{\{\widehat{\mu}_{1:s}(A_{h,j})\geq\log(n)/n\}} - \frac{\int_{A_{h,j}}m_\Delta(x)\mu(dx)}{\mu(A_{h,j})}\right|.$$

For any $j$ with $\widehat{\mu}_{1:s}(A_{h,j})\geq\log(n)/n$ we have

$$\left|\frac{\widehat{\nu}_{1:s}(A_{h,j})}{\widehat{\mu}_{1:s}(A_{h,j})} - \frac{\int_{A_{h,j}}m_\Delta(x)\mu(dx)}{\mu(A_{h,j})}\right|$$

$$\leq |\widehat{\nu}_{1:s}(A_{h,j})|\left|\frac{1}{\widehat{\mu}_{1:s}(A_{h,j})} - \frac{1}{\mu_{1:s}(A_{h,j})}\right| + \frac{|\widehat{\nu}_{1:s}(A_{h,j})-\nu_{1:s}(A_{h,j})|}{\mu_{1:s}(A_{h,j})} + \left|\frac{\nu_{1:s}(A_{h,j})}{\mu_{1:s}(A_{h,j})} - \frac{\int_{A_{h,j}}m_\Delta(x)\mu(dx)}{\mu(A_{h,j})}\right|$$

$$= \left|\frac{\widehat{\nu}_{1:s}(A_{h,j})}{\widehat{\mu}_{1:s}(A_{h,j})}\right|\left|\frac{(4/\alpha)\sum_{i=1}^s\epsilon_{i,j}}{\sum_{i=1}^s\mathbb{1}_{\{X_i\in A_{h,j}\}}}\right| + \left|\frac{(4M/\alpha)\sum_{i=1}^s\zeta_{i,j}}{\sum_{i=1}^s\mathbb{1}_{\{X_i\in A_{h,j}\}}}\right| + \left|\frac{\nu_{1:s}(A_{h,j})}{\mu_{1:s}(A_{h,j})} - \frac{\int_{A_{h,j}}m_\Delta(x)\mu(dx)}{\mu(A_{h,j})}\right|.$$

The final term can be bounded exactly as in the proof of Lemma S.5. For the first two terms

$$\mathbb{P}\left(\left|\frac{\widehat{\nu}_{1:s}(A_{h,j})}{\widehat{\mu}_{1:s}(A_{h,j})}\right|\left|\frac{(4/\alpha)\sum_{i=1}^s\epsilon_{i,j}}{\sum_{i=1}^s\mathbb{1}_{\{X_i\in A_{h,j}\}}}\right| + \left|\frac{(4M/\alpha)\sum_{i=1}^s\zeta_{i,j}}{\sum_{i=1}^s\mathbb{1}_{\{X_i\in A_{h,j}\}}}\right|\geq\eta\right)$$

$$\leq\mathbb{P}\left(\left|\frac{\widehat{\nu}_{1:s}(A_{h,j})}{\widehat{\mu}_{1:s}(A_{h,j})}\right|\left|\frac{(4/\alpha)\sum_{i=1}^s\epsilon_{i,j}}{\sum_{i=1}^s\mathbb{1}_{\{X_i\in A_{h,j}\}}}\right|\geq\eta/2\right) + \mathbb{P}\left(\left|\frac{(4M/\alpha)\sum_{i=1}^s\zeta_{i,j}}{\sum_{i=1}^s\mathbb{1}_{\{X_i\in A_{h,j}\}}}\right|\geq\eta/2\right)$$

$$\leq\mathbb{P}\left(\left|\frac{\widehat{\nu}_{1:s}(A_{h,j})}{\widehat{\mu}_{1:s}(A_{h,j})}\right|\geq 4M\right) + \mathbb{P}\left(\left|\frac{(4/\alpha)\sum_{i=1}^s\epsilon_{i,j}}{\sum_{i=1}^s\mathbb{1}_{\{X_i\in A_{h,j}\}}}\right|\geq\eta/(8M)\right) + \mathbb{P}\left(\left|\frac{(4M/\alpha)\sum_{i=1}^s\zeta_{i,j}}{\sum_{i=1}^s\mathbb{1}_{\{X_i\in A_{h,j}\}}}\right|\geq\eta/2\right)$$

$$\leq\mathbb{P}\left(\left|\frac{\sum_{i=1}^s Y_i\mathbb{1}_{\{X_i\in A_{h,j}\}}+\frac{4M}{\alpha}\sum_{i=1}^s\zeta_{i,j}}{\sum_{i=1}^s\mathbb{1}_{\{X_i\in A_{h,j}\}}+\frac{4}{\alpha}\sum_{i=1}^s\epsilon_{i,j}}\right|\geq 4M\right) + 2\mathbb{P}\left(\left|\frac{\sum_{i=1}^s\epsilon_{i,j}}{\sum_{i=1}^s\mathbb{1}_{\{X_i\in A_{h,j}\}}}\right|\geq\eta\alpha/(32M)\right)$$

$$\leq\mathbb{P}\left(\frac{4}{\alpha}\sum_{i=1}^s\epsilon_{i,j}<-\frac{1}{2}\sum_{i=1}^s\mathbb{1}_{\{X_i\in A_{h,j}\}}\right) + \mathbb{P}\left(\frac{|\sum_{i=1}^s Y_i\mathbb{1}_{\{X_i\in A_{h,j}\}}+\frac{4M}{\alpha}\sum_{i=1}^s\zeta_{i,j}|}{\sum_{i=1}^s\mathbb{1}_{\{X_i\in A_{h,j}\}}}\geq 2M\right)$$

$$+ 2\mathbb{P}\left(\left|\frac{\sum_{i=1}^s\epsilon_{i,j}}{\sum_{i=1}^s\mathbb{1}_{\{X_i\in A_{h,j}\}}}\right|\geq\eta\alpha/(32M)\right)$$

$$\leq 4\mathbb{P}\left(\left|\sum_{i=1}^s\epsilon_{i,j}\right|\geq\alpha\min\{\eta/(32M),1/8\}\sum_{i=1}^s\mathbb{1}_{\{X_i\in A_{h,j}\}}\right)$$

$$\leq 4\mathbb{P}\left(\left|\frac{1}{s}\sum_{i=1}^s\epsilon_{i,j}\right|\geq\alpha\mu(A_{h,j})\min\{\eta/(64M),1/16\}\right) + 4\mathbb{P}\left(\sum_{i=1}^s\mathbb{1}_{\{X_i\in A_{h,j}\}}<s\mu(A_{h,j})/2\right).$$

We have thus reduced our problem to two standard concentration inequalities. By Lemma S.6 we have

$$\mathbb{P}\left(\left|\sum_{i=1}^s\epsilon_{i,j}\right|\geq\alpha\mu(A_{h,j})\min\{\eta/(64M),1/16\}\right)\leq 2\exp\left(-s\alpha^2\mu(A_{h,j})^2\min\{\eta^2/(2^{14}M^2),2^{-6}\}\right).$$

Using the fact that $e^{-x}\leq 1-x+x^2/2$ for all $x>0$ we have the Chernoff bound

$$\mathbb{P}\left(\sum_{i=1}^s\mathbb{1}_{\{X_i\in A_{h,j}\}}<s\mu(A_{h,j})/2\right)\leq\inf_{\lambda>0}e^{\lambda\mu(A_{h,j})/2}\{1-\mu(A_{h,j})+\mu(A_{h,j})e^{-\lambda/s}\}^s$$

$$\leq\inf_{\lambda>0}e^{\lambda\mu(A_{h,j})/2}\{1-\lambda\mu(A_{h,j})/s+\lambda^2\mu(A_{h,j})/(2s^2)\}^s$$

$$\leq\inf_{\lambda>0}\exp\left(-\lambda\mu(A_{h,j})/2+\lambda^2\mu(A_{h,j})/(2s)\right)$$

$$= \exp\left(-\frac{s\mu(A_{h,j})}{8}\right).$$

Hence,

$$\mathbb{P}\left(\left|\frac{\widehat{\nu}_{1:s}(A_{h,j})}{\widehat{\mu}_{1:s}(A_{h,j})}\right|\left|\frac{(4/\alpha)\sum_{i=1}^{s}\epsilon_{i,j}}{\sum_{i=1}^{s}\mathbb{1}_{\{X_i\in A_{h,j}\}}}\right|+\left|\frac{(4M/\alpha)\sum_{i=1}^{s}\zeta_{i,j}}{\sum_{i=1}^{s}\mathbb{1}_{\{X_i\in A_{h,j}\}}}\right|\geq\epsilon\right)$$
$$\leq 16\exp\left(-s\alpha^2\mu(A_{h,j})^2\min\{\epsilon^2/(2^{14}M^2),2^{-6}\}\right).$$

Combining the previous bound with bounds from the proof of Lemma S.5, when $\eta\in(0,2^6M\sqrt{s(t-s)/t}]$ we have

$$\mathbb{P}\left(\sqrt{\frac{s(t-s)}{t}}\left|\sup_{x\in\mathbb{R}^d}|\widehat{m}_{1:s}(x)-\widehat{m}_{(s+1):t}(x)|-\sup_{x\in\mathbb{R}^d}|m_{1:s}^{(h)}(x)-m_{(s+1):t}^{(h)}(x)|\right|\geq\eta+2\sqrt{\frac{s(t-s)}{t}}C_{\mathrm{Lip}}\sqrt{d}h\right)$$

$$\leq\mathbb{P}\left(\sqrt{\frac{s(t-s)}{t}}\sup_x|\widehat{m}_{1:s}(x)-m_{1:s}^{(h)}(x)|\geq\eta/4+\sqrt{\frac{s(t-s)}{t}}C_{\mathrm{Lip}}\sqrt{d}h\right)$$

$$+\mathbb{P}\left(\sqrt{\frac{s(t-s)}{t}}\sup_x|\widehat{m}_{(s+1):t}(x)-m_{(s+1):t}^{(h)}(x)|\geq\eta/4+\sqrt{\frac{s(t-s)}{t}}C_{\mathrm{Lip}}\sqrt{d}h\right)$$

$$+\mathbb{P}\left(\sqrt{\frac{s(t-s)}{t}}\max_j\left\{\left|\frac{\widehat{\nu}_{1:s}(A_{h,j})}{\widehat{\mu}_{1:s}(A_{h,j})}\right|\left|\frac{(4/\alpha)\sum_{i=1}^{s}\epsilon_{i,j}}{\sum_{i=1}^{s}\mathbb{1}_{\{X_i\in A_{h,j}\}}}\right|+\left|\frac{(4M/\alpha)\sum_{i=1}^{s}\zeta_{i,j}}{\sum_{i=1}^{s}\mathbb{1}_{\{X_i\in A_{h,j}\}}}\right|\right\}\geq\eta/4\right)$$

$$+\mathbb{P}\left(\sqrt{\frac{s(t-s)}{t}}\max_j\left\{\left|\frac{\widehat{\nu}_{(s+1):t}(A_{h,j})}{\widehat{\mu}_{(s+1):t}(A_{h,j})}\right|\left|\frac{(4/\alpha)\sum_{i=s+1}^{t}\epsilon_{i,j}}{\sum_{i=s+1}^{t}\mathbb{1}_{\{X_i\in A_{h,j}\}}}\right|+\left|\frac{(4M/\alpha)\sum_{i=s+1}^{t}\zeta_{i,j}}{\sum_{i=s+1}^{t}\mathbb{1}_{\{X_i\in A_{h,j}\}}}\right|\right\}\geq\eta/4\right)$$

$$\leq\frac{2}{c_{\min}h^d}\left\{\exp\left(-\frac{t}{t-s}\frac{c_{\min}h^d\eta^2}{64M^2}\right)+\exp\left(-\frac{t}{s}\frac{c_{\min}h^d\eta^2}{64M^2}\right)+8\exp\left(-s\alpha^2c_{\min}^2h^{2d}\min\left\{\frac{t\eta^2}{s(t-s)2^{18}M^2},2^{-6}\right\}\right)\right.$$

$$\left.+8\exp\left(-(t-s)\alpha^2c_{\min}^2h^{2d}\min\left\{\frac{t\eta^2}{s(t-s)2^{18}M^2},2^{-6}\right\}\right)\right\}$$

$$\leq\frac{4}{c_{\min}h^d}\left\{\exp\left(-\frac{c_{\min}h^d\eta^2}{64M^2}\right)+8\exp\left(-\frac{\alpha^2c_{\min}^2h^{2d}\eta^2}{2^{18}M^2}\right)\right\}\leq\frac{36}{c_{\min}h^d}\exp\left(-\frac{\alpha^2c_{\min}^2h^{2d}\eta^2}{2^{18}M^2}\right).$$

The result follows on taking

$$\eta=\frac{2^9M}{c_{\min}h^d\alpha}\sqrt{\log(72t^3)+\log(1/c_{\min})+\log(h^{-d})+\log(1/\gamma)}.$$

$\square$

## S.3  Proofs of the results in Appendix A

**Lemma S.8.** *For any $\gamma>0$, it holds that*

$$\mathbb{P}\left\{\exists s,t\in\mathbb{N},\,t>1,\,s\in[1,t):\left|\left(\frac{t-s}{ts}\right)^{1/2}\sum_{l=1}^{s}(Z_l-f_l)-\left\{\frac{s}{t(t-s)}\right\}^{1/2}\sum_{l=s+1}^{t}(Z_l-f_l)\right|\right.$$

$$\left.>2^{3/2}\sqrt{\sigma^2+4\alpha^{-2}}\log^{1/2}(t/\gamma)\right\}\leq\gamma.$$

*Proof.* It holds that for any sequence $\{\varepsilon_t>0\}$,

$$\mathbb{P}\left\{\exists s,t\in N,\,t>1,\,s\in[1,t):\left|\left(\frac{t-s}{ts}\right)^{1/2}\sum_{l=1}^{s}(Z_l-f_l)-\left\{\frac{s}{t(t-s)}\right\}^{1/2}\sum_{l=s+1}^{t}(Z_l-f_l)\right|\geq\varepsilon_t\right\}$$

$$\leq\sum_{j=1}^{\infty}\mathbb{P}\left\{\max_{2^j\leq t<2^{j+1}}\max_{1\leq s<t}\left|\left(\frac{t-s}{ts}\right)^{1/2}\sum_{l=1}^{s}(Z_l-f_l)-\left\{\frac{s}{t(t-s)}\right\}^{1/2}\sum_{l=s+1}^{t}(Z_l-f_l)\right|\geq\varepsilon_t\right\}$$

$$\leq \sum_{j=1}^{\infty} 2^j \max_{2^j \leq t < 2^{j+1}} \mathbb{P} \left\{ \max_{1 \leq s < t} \left| \left( \frac{t-s}{ts} \right)^{1/2} \sum_{l=1}^{s} (Z_l - f_l) - \left\{ \frac{s}{t(t-s)} \right\}^{1/2} \sum_{l=s+1}^{t} (Z_l - f_l) \right| \geq \varepsilon_t \right\}$$

$$\leq \sum_{j=1}^{\infty} 2^j \max_{2^j \leq t < 2^{j+1}} t \mathbb{P} \left\{ |W| \geq \varepsilon_t \right\} \leq \sum_{j=1}^{\infty} 2^{2j+1} \mathbb{P} \left\{ |W| \geq \varepsilon_j' \right\},$$

where $\varepsilon_t = \varepsilon_j'$, for any $t \in \{2^j, \ldots, 2^{j+1} - 1\}$, $j = 1, 2, \ldots$ and $W$ is a mean zero sub-Gaussian random variable with

$$\|W\|_{\psi_2} \leq \sqrt{\sigma^2 + 4\alpha^{-2}}. \tag{S.17}$$

We remark that (S.17) is due to Assumption 6, which implies that $\mathbb{E}(Z_l) = f_l$ and

$$\|Z_l - f_l\|_{\psi_2} \leq \sqrt{\|X_l - f_l\|_{\psi_2}^2 + \|\epsilon_l\|_{\psi_2}^2} = \sqrt{\sigma^2 + 4\alpha^{-2}},$$

where the last identity follows from Lemma 1 in [1].

For any $t = 1, 2, \ldots$, let

$$\varepsilon_t = \sqrt{2\sigma^2 + 8\alpha^{-2}} \left[ 2\log(t) + \log\log(t) + \log\{\log(t) + \log(2)\} - 2\log\log(2) - \log(\gamma) \right]^{1/2}.$$

Due to the sub-Gaussianity, we have that for any $\zeta > 0$, $\mathbb{P}\{|W| \geq \zeta\} < 2\exp(-2^{-1}\zeta^2/(\sigma^2 + 4\alpha^{-2}))$ [e.g. (2.9) in 3].

$$\mathbb{P} \left\{ \exists s, t \in N, \ t > 1, \ s \in [1, t) : \left| \left( \frac{t-s}{ts} \right)^{1/2} \sum_{l=1}^{s} (Z_l - f_l) - \left\{ \frac{s}{t(t-s)} \right\}^{1/2} \sum_{l=s+1}^{t} (Z_l - f_l) \right| \geq \varepsilon_t \right\}$$

$$\leq \sum_{j=1}^{\infty} \max_{2^j \leq t \leq 2^{j+1}} \exp[(2j+2)\log(2) - 2\log(t) - \log\log(t) - \log\{\log(t) + \log(2)\} + 2\log\log(2) + \log(\gamma)]$$

$$\leq \sum_{j=1}^{\infty} \exp \left\{ (2j+2)\log(2) - 2j\log(2) - 2\log(j) - \log\{(j+1)\log(2)\} + 2\log\log(2) + \log(\gamma) \right\}$$

$$\leq \gamma \sum_{j=1}^{\infty} \frac{1}{j(j+1)} \leq \gamma.$$

For simplicity, we let

$$\varepsilon_t = 2^{3/2} \sqrt{\sigma^2 + 4\alpha^{-2}} \log^{1/2}(t/\gamma),$$

which satisfies that for any $t \geq 2$ and $\gamma \in (0, 1)$,

$$2^{3/2} \log^{1/2}(t/\gamma) \geq \sqrt{2} [2\log(t) + \log\log(t) + \log\{\log(t) + \log(2)\} - 2\log\log(2) - \log(\gamma)]^{1/2}. \tag{S.18}$$

We therefore completes the proof. $\qquad\square$

*Proof of Theorem 5.* **Step 1.** Define the event

$$\mathcal{B} = \left\{ \forall s, t \in N, \ t > 1, \ s \in [1, t) : \left| \left( \frac{t-s}{ts} \right)^{1/2} \sum_{l=1}^{s} (Z_l - f_l) - \left\{ \frac{s}{t(t-s)} \right\}^{1/2} \sum_{l=s+1}^{t} (Z_l - f_l) \right| < b_t \right\}. \tag{S.19}$$

It follows from Lemma S.8 that $\mathbb{P}\{\mathcal{B}\} > 1 - \gamma$. Throughout the proof we assume that the event $\mathcal{B}$ holds.

For any $s, t \in \mathbb{N}$, $1 \leq s < t$, it holds that $\left| \widehat{D}_{s,t} - D_{s,t} \right| < b_t$, which implies that

$$D_{s,t} + b_t > \widehat{D}_{s,t} > D_{s,t} - b_t. \tag{S.20}$$

**Step 2.** For any $t \leq \Delta$, we have that $D_{s,t} = 0$, for all $s \in [1, t)$. Thus, using (S.20), we conclude that, $\widehat{t} > t$ and, therefore that $\widehat{t} > \Delta$.

**Step 3.** Now we consider any $t > \Delta$. If there exists $s \in [1, t)$ such that $\widehat{D}_{s,t} > b_t$, then $d \leq t - \Delta$. Thus, $d \leq \widetilde{t} - \Delta$, where

$$\widetilde{t} = \min\{t > \Delta, \exists s \in [\Delta, t), \widehat{D}_{s,t} > b_t\},$$

and any upper bound on $\widetilde{t}$ will also be an upper bound on $d$, when the signal-to-noise constraint specified by Assumption 8 is satisfied. Thus, our task becomes that of computing a sharp upper bound on $\widetilde{t}$. To that effect, notice that, when $\Delta \leq s < t$,

$$D_{s,t} = \Delta \left(\frac{t-s}{ts}\right)^{1/2} |\mu_1 - \mu_2| = \Delta \left(\frac{t-s}{ts}\right)^{1/2} \kappa, \tag{S.21}$$

and, because of (S.20) again,

$$\widehat{D}_{s,t} \geq \Delta \left(\frac{t-s}{ts}\right)^{1/2} \kappa - b_t.$$

As a result, we obtain that $\widetilde{t} \leq t^*$, where

$$t_* = \min \left\{ t > \Delta : \max_{s \in [\Delta, t)} \left\{ \Delta \left(\frac{t-s}{ts}\right)^{1/2} \kappa - 2b_t \right\} \geq 0 \right\}.$$

**Step 4.** Write for convenience $m = t^* - \Delta$, so that $\widehat{t} - \Delta \leq m$. Recalling that

$$b_t = 2^{3/2} \sqrt{\sigma^2 + 4\alpha^{-2}} \log^{1/2}(t/\gamma),$$

we seek the smallest integer $m$ such that

$$\max_{s \in [\Delta, m+\Delta)} \left[ \Delta \kappa \left\{ \frac{m + \Delta - s}{(m+\Delta)s} \right\}^{1/2} - 2^{5/2} \sqrt{\sigma^2 + 4\alpha^{-2}} \log^{1/2}\{(m+\Delta)/\gamma\} \right] > 0,$$

which is equivalent to finding the smallest integer $m$ such that

$$\max_{s \in [\Delta, m+\Delta)} \left[ \Delta^2 \kappa^2 - 32(\sigma^2 + 4\alpha^{-2}) \frac{s(m+\Delta)}{m+\Delta-s} \log\{(m+\Delta)/\gamma\} \right] > 0.$$

In turn, the above task corresponds to that of computing the smallest integer $m$ such that

$$\Delta^2 \frac{\kappa^2}{\sigma^2 + 4\alpha^{-2}} > \min_{s \in [\Delta, m+\Delta)} \left[ 32 \frac{s(m+\Delta)}{m+\Delta-s} \log\{(m+\Delta)/\gamma\} \right]$$

$$= 32 \frac{\Delta(m+\Delta)}{m} \log\{(m+\Delta)/\gamma\},$$

or, equivalently, such that

$$m \left[ \frac{\Delta \kappa^2}{32(\sigma^2 + 4\alpha^{-2})} - \log\{(m+\Delta)/\gamma\} \right] > \Delta \log\{(m+\Delta)/\gamma\}, \tag{S.22}$$

under Assumption 8.

Let $C_d$ be an absolute constant large enough and also upper bounded by $C_{\text{SNR}}$. The claimed result now follows once we show that that the value

$$m^* = \lceil C_d \log(\Delta/\gamma)(\sigma^2 + 4\alpha^{-2})\kappa^{-2} \rceil$$

satisfies (S.22). To see this, assume for simplicity that $C_d \log(\Delta/\gamma)(\sigma^2 + 4\alpha^{-2})\kappa^{-2}$ is an integer; if not, the proof only requires trivial modifications. We first point out that $m^* \leq \Delta$ because of Assumption 8 and the fact that $C_d \leq C_{\text{SNR}}$. Now, the left hand side of inequality (S.22) is equal, for this choice of $m$, to

$$C_d \log(\Delta/\gamma)\frac{\Delta}{32} - C_d \frac{\sigma^2 + 4\alpha^{-2}}{\kappa^2} \log(\Delta/\gamma) \log \left\{ \frac{C_d \log(\Delta/\gamma)(\sigma^2 + 4\alpha^{-2})/\kappa^2 + \Delta}{\gamma} \right\}. \tag{S.23}$$

Using again Assumption 8 and the fact that $m^* \leq \Delta$, the second term in the previous expression is upper bounded by

$$\frac{2C_d}{C_{\text{SNR}}} \Delta \log(\Delta/\gamma),$$

due to the fact that $2\log(x) \geq \log(2x)$, $x \geq 2$. Thus, the quantity in (S.23) is lower bounded by

$$\Delta \log(\Delta/\gamma) \left( \frac{C_d}{32} - \frac{2C_d}{C_{\mathrm{SNR}}} \right) \geq 2\Delta \log(\Delta/\gamma) \geq \Delta \log(2\Delta/\gamma) \geq \Delta \log\left((m^* + \Delta)/\gamma\right),$$

where the first inequality is justified by first choosing a large enough $C_d$ and then choosing $C_{\mathrm{SNR}}$ larger than $C_d$, and the second and third inequalities follow from $\log(\Delta/\gamma) \geq 0$ and $m^* \leq \Delta$, respectively. Thus, combining the last display with (S.22) and (S.23) yields (16). Finally, (14) and (15) follow immediately from Steps 1 and 2. $\qquad\square$

*Proof of Theorem 6.* Throughout the proof we will assume for simplicity that $c\sigma^2\alpha^{-2}\log(1/\gamma)\kappa^{-2}$ is an integer.

**Step 1.** For any $n$, let $P^n$ be the restrictions of a distribution $P$ to $\mathcal{F}_n$, i.e. the $\sigma$-field generated by the observations $\{Z_i\}_{i=1}^n$.

Let $P_1$ be the distribution of the original $X_t$ before the change and let $P_2$ be the distribution after. Write

$$m_{t,k}(z|z_1, \ldots, z_{t-1}) = \int q_t(z|x, z_1, \ldots, z_{t-1}) \, dP_k(x)$$

for the conditional density of the $Z_t$ before $(k = 1)$ and after $(k = 2)$ the change point. To be specific, we let

$$P_1 = \mathrm{Unif}[0, 2\sigma], \quad P_2 = \kappa + \mathrm{Unif}[0, 2\sigma],$$

which satisfy that

$$\|P_1 - P_2\|_{\mathrm{TV}} = \frac{\kappa}{2\sigma}, \tag{S.24}$$

assuming that $\kappa < 2\sigma$.

For any $\nu \geq 1$ and $n \geq \nu$, we have that for any $n \geq \Delta$, it holds that

$$\frac{dP_{\kappa,\sigma,\nu}^n}{dP_{\kappa,\sigma,\infty}^n} = \exp\left( \sum_{i=\nu+1}^n W_i \right),$$

where $P_{\kappa,\sigma,\infty}$ indicates the joint distribution under which there is no change point and

$$W_i = \log \frac{m_{i,2}(Z_i|Z_{i-1}, \ldots, Z_1)}{m_{i,1}(Z_i|Z_{i-1}, \ldots, Z_1)}.$$

For any $\nu \geq 1$, define the event

$$\mathcal{E}_\nu = \left\{ \nu < T < \nu + \frac{c\sigma^2\alpha^{-2}\log(1/\gamma)}{\kappa^2}, \ \sum_{i=\nu+1}^T W_i < \frac{3}{4}\log\left(\frac{1}{\alpha}\right) \right\}.$$

Then we have

$$\mathbb{P}_{\kappa,\sigma,\nu}(\mathcal{E}_\nu) = \int_{\mathcal{E}_\nu} \exp\left( \sum_{i=\nu+1}^T W_i \right) dP_{\kappa,\sigma,\infty} \leq \exp\left\{ (3/4)\log(1/\gamma) \right\} \mathbb{P}_{\kappa,\sigma,\infty}(\mathcal{E}_\nu)$$

$$\leq \exp\left\{ (3/4)\log(1/\gamma) \right\} \mathbb{P}_{\kappa,\sigma,\infty}\left\{ \nu < T < \nu + \frac{c\sigma^2\alpha^{-2}\log(1/\gamma)}{\kappa^2} \right\} \leq \gamma^{-3/4}\gamma = \gamma^{1/4}, \quad \text{(S.25)}$$

where the first two inequalities follow from the definition of $\mathcal{E}_\nu$, and the last inequality follows from the definition of $\mathcal{D}(\gamma)$.

**Step 2.** Note that for any $i, z, z_{i-1}, \ldots, z_1$ and an arbitrary $x_0$ we have

$$\frac{m_{i,2}(z|z_{i-1}, \ldots, z_1)}{m_{i,1}(z|z_{i-1}, \ldots, z_1)} = \frac{\int q_{i,1}(z|x, z_{i-1}, \ldots, z_1) \, dP_2(x)}{\int q_{i,2}(z|x, z_{i-1}, \ldots, z_1) \, dP_1(x)} \leq \frac{e^\alpha q(z|x_0, z_{i-1}, \ldots, z_1) \int dP_2(x)}{e^{-\alpha} q(z|x_0, z_{i-1}, \ldots, z_1) \int dP_1(x)} = e^{2\alpha},$$

and we can similarly see that $m_{i,2}(z|z_{i-1}, \ldots, z_1)/m_{i,1}(z|z_{i-1}, \ldots, z_1) \geq e^{-2\alpha}$. When $\alpha$ is small, therefore, (say $\alpha \leq 0.5\log(0.5)$), we will have for any $z_1, \ldots, z_{i-1}$ that

$$\int m_{i,2}(z|z_{i-1}, \ldots, z_1) \log \frac{m_{i,2}(z|z_{i-1}, \ldots, z_1)}{m_{i,1}(z|z_{i-1}, \ldots, z_1)} \, dz$$

$$\leq -\int m_{i,2}(z|z_{i-1},\ldots,z_1)\left\{\frac{m_{i,1}(z|z_{i-1},\ldots,z_1)}{m_{i,2}(z|z_{i-1},\ldots,z_1)}-1-\left(\frac{m_{i,1}(z|z_{i-1},\ldots,z_1)}{m_{i,2}(z|z_{i-1},\ldots,z_1)}-1\right)^2\right\}dz$$

$$=\int \frac{\{m_{i,2}(z|z_{i-1},\ldots,z_1)-m_{i,1}(z|z_{i-1},\ldots,z_1)\}^2}{m_{i,2}(z|z_{i-1},\ldots,z_1)}\,dz \leq \min(4,e^{2\alpha})(e^\alpha-1)^2\|P_1-P_2\|_{\mathrm{TV}}^2,$$

where the final inequality is Lemma 1 in [2, Supplementary material]. Calculations around Lemma 1 in [2, Supplementary material] also reveal that

$$\left|\log\frac{m_{i,2}(z|z_{i-1},\ldots,z_1)}{m_{i,1}(z|z_{i-1},\ldots,z_1)}\right| \leq \min(2,e^\alpha)(e^\alpha-1)\|P_1-P_2\|_{\mathrm{TV}}.$$

It follows from the Azuma–Hoeffding inequality [e.g. 3, Corollary 2.20] that for any $x>0$ and $t\in\mathbb{N}$ we have

$$\mathbb{P}\left(\sum_{i=\nu+1}^{\nu+t}W_i \geq x+t\min(4,e^{2\alpha})(e^\alpha-1)^2\|P_1-P_2\|_{\mathrm{TV}}^2 \ \Big| \ Z_\nu,\ldots,Z_1\right)$$

$$\leq \mathbb{P}\left(\sum_{i=\nu+1}^{\nu+t}\{W_i-\mathbb{E}(W_i|Z_{i-1},\ldots,Z_1)\} \geq x \ \Big| \ Z_\nu,\ldots,Z_1\right)$$

$$\leq \exp\left(-\frac{2x^2}{t\min(4,e^{2\alpha})(e^\alpha-1)^2\|P_1-P_2\|_{\mathrm{TV}}^2}\right)$$

Due to (S.24) and our assumption that $\alpha \leq 1$, for small enough $c>0$ we have that

$$\frac{c\sigma^2\alpha^{-2}\log(1/\gamma)}{\kappa^2}\times \min(4,e^{2\alpha})(e^\alpha-1)^2\|P_1-P_2\|_{\mathrm{TV}}^2 \leq \frac{1}{4}\log(1/\gamma).$$

For any $\nu\geq 1$ and $T\in\mathcal{D}(\gamma)$, since $\{T\geq\nu\}\in\mathcal{F}_{\nu-1}$, it therefore follows that for such $c$ we have

$$\mathbb{P}_{\kappa,\sigma,\nu}\left\{\nu<T<\nu+\frac{c\sigma^2\alpha^{-2}\log(1/\gamma)}{\kappa^2},\ \sum_{i=\nu+1}^T W_i \geq (3/4)\log(1/\gamma)\ \Big|\ T>\nu\right\}$$

$$\leq \operatorname{ess\,sup}\mathbb{P}_{\kappa,\sigma,\nu}\left\{\max_{1\leq t\leq \frac{c\sigma^2\alpha^{-2}\log(1/\gamma)}{\kappa^2}}\sum_{i=\nu+1}^{\nu+t}W_i \geq (3/4)\log(1/\gamma)\ \Big|\ Z_1,\ldots,Z_\nu\right\}$$

$$\leq \operatorname{ess\,sup}\mathbb{P}_{\kappa,\sigma,\nu}\left[\max_{1\leq t\leq \frac{c\sigma^2\alpha^{-2}\log(1/\gamma)}{\kappa^2}}\sum_{i=\nu+1}^{\nu+t}\{W_i-\mathbb{E}(W_i|Z_{i-1},\ldots,Z_1)\} \geq (1/2)\log(1/\gamma)\Big|Z_1,\ldots,Z_\nu\right]$$

$$\leq \frac{c\sigma^2\alpha^{-2}\log(1/\gamma)}{\kappa^2}\exp\left\{-\frac{(1/2)\log^2(1/\gamma)}{\frac{c\sigma^2\alpha^{-2}\log(1/\gamma)}{\kappa^2}\min(4,e^{2\alpha})(e^\alpha-1)^2\|P_1-P_2\|_{\mathrm{TV}}^2}\right\}$$

$$\leq \frac{c\sigma^2\alpha^{-2}\log(1/\gamma)}{\kappa^2}\exp\left\{-\log(1/\gamma)\right\} \leq \gamma^{1/4},$$

where the third inequality follows by a union bound argument, the fourth inequality holds for small enough $c>0$, and the last inequality holds for small enough $\gamma$. Since the upper bound is independent of $\nu$, it holds that

$$\sup_{\nu\geq 1}\mathbb{P}_{\kappa,\sigma,\nu}\left\{\nu<T<\nu+\frac{c\sigma^2\alpha^{-2}\log(1/\gamma)}{\kappa^2},\ \sum_{i=\nu+1}^T W_i \geq (3/4)\log(1/\gamma)\ \Big|\ T\geq\nu\right\} \leq \gamma^{1/4},$$

which leads to

$$\sup_{\nu\geq 1}\mathbb{P}_{\kappa,\sigma,\nu}\left\{\nu<T<\nu+\frac{c\sigma^2\alpha^{-2}\log(1/\gamma)}{\kappa^2},\ \sum_{i=\nu+1}^T W_i \geq (3/4)\log(1/\gamma)\right\} \leq \gamma^{1/4}. \qquad \text{(S.26)}$$

Combining (S.25) and (S.26), we have

$$\sup_{\nu\geq 1}\mathbb{P}_{\kappa,\sigma,\nu}\left\{\nu<T<\nu+\frac{c\sigma^2\alpha^{-2}\log(1/\gamma)}{\kappa^2}\right\} \leq 2\alpha^{1/4}. \qquad \text{(S.27)}$$

**Step 3.** We now have, for any change point time $\Delta$,

$$
\mathbb{E}_{\kappa,\sigma,\Delta} \left\{ (T - \Delta)_+ \right\} \geq \frac{c\sigma^2 \alpha^{-2} \log(1/\gamma)}{\kappa^2} \mathbb{P}_{\kappa,\sigma,\Delta} \left\{ T - \Delta \geq \frac{c\sigma^2 \alpha^{-2} \log(1/\gamma)}{\kappa^2} \right\}
$$

$$
= \frac{c\sigma^2 \alpha^{-2} \log(1/\gamma)}{\kappa^2} \left[ \mathbb{P}_{\kappa,\sigma,\Delta} \{ T > \Delta \} - \mathbb{P}_{\kappa,\sigma,\Delta} \left\{ \Delta < T < \Delta + \frac{c\sigma^2 \alpha^{-2} \log(1/\gamma)}{\kappa^2} \right\} \right]
$$

$$
\geq \frac{c\sigma^2 \alpha^{-2} \log(1/\gamma)}{2\kappa^2},
$$

where the first inequality is due to Markov's inequality, the second is due to (S.27) and the definition of the class of $\mathcal{D}(\gamma)$ of stopping times. $\qquad \square$