# OpenReview forum: "Locally private online change point detection"
_NeurIPS.cc/2021/Conference — NeurIPS 2021 Poster_

### Official Review · Reviewer_djMH · 2021-07-13

**Rating:** 6
**Confidence:** 2

**Summary:**

Paper proposes an online change point detection method with local differential privacy

**Limitations And Societal Impact:**

Yes

**Main Review:**

An online changepoint detection method is proposed that preserves privacy via the use of differential privacy. Paper is clearly written, but is theoretically dense (which, by no means is a negative), hence making  a fair review in time constrained fashion hard. My views are based on the superficial skimming of the proofs.

The main idea of the paper can be reduced to using a local differentially private chanel to privatize incoming data and then running an online change point detection on it. It is an interesting approach where the argument for non-interactive queries is made. I'm curious to what authors think about the use of other DP data encoding techniques for this approach (from a global perspective), such as if we have some prior data available, cant we train a generative model with DP and then use it subsequently for encoding incoming observations? This might lead to a different set of assumptions, but i think it is possible.

As the paper doesn't have any experiments, it makes it a bit hard to empirically  evaluate the efficacy of the proposed method. If space was the issue, I think the Section 3.3 can be moved into the supplementary material and some empirical results can really strengthen the paper.


I have read authors' response.

**Time Spent Reviewing:**

2

---

> ### Author Response · Authors · 2021-08-10
> **Responses to the comments on numerical results and other methods.**
>
> Thank you for your appreciation and constructive comments. In the following, we response to your comments point-by-point.
>
> * It is indeed intriguing to use prior data if available.  In view of the lower bound results we obtain in Theorem 2, our current result is already optimal up to logarithmic factors.  With a more refined method and prior data, one may further improve the result by up to logarithmic or constant terms.
>
> * We agree that some numerical examples would be necessary and we shall add them in the revision.

---

### Official Review · Reviewer_KJKZ · 2021-07-14

**Rating:** 5
**Confidence:** 3

**Summary:**

The paper studies change point detection for regression under the constraint of local differential privacy (LDP). Its main contribution is a non-interactive (in the sense that all randomizers are chosen ahead of time) algorithm and upper bound on the error of the identified change point, with a sequentially interactive lower bound matching up to logarithmic factors in the failure probability. The paper also provides a non-private benchmark, which is apparently also original.

**Limitations And Societal Impact:**

Some comments on limitations appear above. The paper rejects out of hand the notion that it might have negative societal impact. But to be fair, change point detection is a pretty abstract problem.

**Main Review:**

—Originality—
I’m not aware of previous work studying LDP solutions to change point detection for regression. If I’m following the paper’s presentation correctly, the given algorithm is really a LDP twist on existing methods for change point detection, although the dependence that LDP forces on additional parameters makes the analysis more challenging.

I know there has been some work on tracking changing data over time with LDP (https://arxiv.org/abs/1802.07128, https://arxiv.org/abs/1811.12469). It’s not directly relevant since users in those papers draw new samples/have new data at each time step, but it’s relevant enough to be discussed — I was initially wondering how this paper differed from those.

—Quality—
I didn’t read the proofs, but the claims are believable, and all the material needed to verify the claims seems to be there. Nearly-matching upper and lower bounds are a nice, relatively complete story.

I did want to see more discussion of the storage/runtime cost of the algorithm — if I understand correctly, N_h is exponential in the dimension d. This seems like a nontrivial obstacle to practicality. Am I missing some reason that this is not actually a problem?

—Clarity—
In general, the writing in the submission is clear. I don’t have any concrete suggestions for improving the writing itself. I do think it’s weird that the paper includes a self-contained treatment of a different problem at the beginning of the supplement. I’m not sure what the point of that section is supposed to be.

—Significance—
I think significance is the trickiest question to review. On one hand, nearly-matching upper and lower bounds are a nice contribution, and I don’t think the algorithm and analysis are obvious (at least as somebody unfamiliar with change point detection as its own literature).

However, the upper bound seems to rely *heavily* on auxiliary knowledge of the problem itself. For example, in Theorem 1, M_0, sigma, C_Lip, and kappa are all assumed to be known to the algorithm operator. This seems like a strong (and unrealistic) requirement. I understand this is probably the price of nearly matching upper and lower bounds, but it does curtail any claim the paper has to a practical solution (plus, the N_h issue mentioned above). Possible ways to mitigate this are some discussion of how flexible the utility is wrt exact knowledge of these quantities, or experiments (though that seems like too much to ask at this point).

Overall, my take on the paper is that it provides a good amount of new info on LDP regression change point detection, but that little of this info is practical, or tells us much about LDP itself (there are already many problems where the optimal LDP algorithm is non-interactive, even against fully interactive algorithms). That makes it hard for me to recommend this paper for acceptance. I’ll try to keep an open mind in discussion, since this is kind of subjective, but I have trouble getting excited about papers that provide (what I see as) impractical solutions to new specific problems.

I may also be undervaluing the non-DP result in this paper, since I really can’t assess its significance to the change point literature in general.

UPDATE AFTER READING AUTHORS' RESPONSE: My overall impression is about the same. The assumptions on knowing many parameters are strong (the authors' claim that these can be obtained by "data-driven" methods seems odd in the context of local privacy -- you can get the problem parameters fine, but for some reason change point detection must be done privately?), and the result is a solution with limited practicality. Since I don't think it tells us anything new conceptually about differential privacy, and it doesn't give us tools that we might use in the real world, I suggest rejecting the current version of the paper.

**Time Spent Reviewing:**

4

---

> ### Author Response · Authors · 2021-08-10
> **Response to the comments on the comparisons with other literature, computational costs and presentation.**
>
> Thank you for your appreciation and constructive comments. In the following, we response to your comments point-by-point.
>
> * __Originality.__  Thank you very much for bringing the references up to us.  We will add them in the revision.  Besides the model differences, the paper (https://arxiv.org/abs/1802.07128) studies an evolving model, where the potential changes happen at fixed time points, while we focus on estimating where the distributional changes occur.  The paper https://arxiv.org/abs/1811.12469 considers a fixed time horizon with diverging number of observations at each time point, while in our paper, at every time point only one observation is available and the number of time points is diverging.  More importantly, the observations in this paper are only taking values in $\{\pm 1, 0\}$, while in our case, we consider multivariate nonparametric regression data.
>
> * __Quality.__  You are indeed right that $N_h$ is exponential in the dimension $d$.  In this paper, we consider $d$ to be fixed but arbitrary.  In fact, the localisation error rate also deteriorates exponentially in $d$, so more tailored algorithms are necessary when $d$ is moderately large.  Our main algorithm is Algorithm 1.  Obtaining a new data point at time point $t$, the added computational cost is of order $O(t \times \mathrm{Cost})$, where Cost is the computational cost of calculating $\widehat{D}_{s, t}$, for any $s \leq t$.  This can be improved by adopting a dyadic grid search (as mentioned in line 117) to $O(\log(t) \times \mathrm{Cost})$.  Regarding the Cost, there exists a tradeoff between the computational cost and the storage cost.  We present two extreme cases: with the storage cost of $O(t)$, $\mathrm{Cost} = O(N_h)$; with the storage cost of $O(1)$, $\mathrm{Cost} = O(t N_h)$.
>
> * __Clarity.__  The univariate case studied at the beginning of the supplementary materials serves as the blueprint and benchmark for understanding more complex data types, including the nonparametric regression problem studied in the main text of our paper.  We are indeed grateful for you to bring this up and we shall improve the presentation of our paper in the revision, by presenting the key ideas of the univariate case in the main text within the paper limit.
>
> * __Significance.__  You are indeed right that the specific requirements on all the tuning parameters are necessary for the theoretical optimality.  In practice, it is common to adopt data-driven methods to choose tuning parameters.  We agree that some numerical examples would be necessary to illustrate how to do this, and we shall add them in the revision.

---

> ### Author Response · Authors · 2021-08-31
> **Responses to the updates**
>
> We would like to thank the reviewer for considering our rebuttal and engaging with the discussion.
>
> We agree that the need to choose tuning parameters is a drawback of our method, but this is a ubiquitous problem in both nonparametric statistics and all online change point detection problems. The focus of our work was on deriving the information theoretic optimal detection delay for this problem, given standard smoothness and tail conditions, and thus quantifying the cost of privacy in the detection delay. As far as we are aware, this is the first work on locally private change point detection, which is an important problem in its own right. The derivation of optimal adaptive methodology without tuning parameters is very difficult, particularly in LDP settings, and we felt that this was beyond the scope of this first work in the area. Nevertheless, we do have heuristics on tuning parameter selection and the practical implementation of our procedures which can be included in a revision.
>
> It is true that $h$ and $M$ must be chosen before any data is collected but, given these, we can make heuristic choices of the thresholds $b_{s,t}$ in a data-driven way using the first portion of the privatised data stream as a training data set. We may permute the ordering of this stream some number $B$ times and choose a fixed threshold such that our procedure declares a change point in $\gamma B$ of the permuted streams. Note also that, if the responses are bounded, as is often assumed in the LDP literature, then $M$ can be chosen equal to this bound and ignored. To choose the bandwidth $h$ for optimal power something must be known about the smoothness of the regression functions, but for any choice of $h$ the above method will result in an approximately valid (controlling false alarm probability) procedure. We can give further detail on this practical approach and illustrate it with simulations in any revision of our work.

---

### Official Review · Reviewer_ALYo · 2021-07-14

**Rating:** 7
**Confidence:** 4

**Summary:**

This paper studies the online CPD problem under the constraint of LDP. In particular, they develop an algorithm to detect changes in the regression function between pairs (X, Y).

**Limitations And Societal Impact:**

Yes

**Main Review:**

This is a high quality paper with important contributions to the CPD research field. Let me summarize some comments:

1. Significance: This paper makes significant contributions in

(1) Proposing the new setting where they focus on online CPD in a more strict version of local privacy constraints. This is a realistic but currently under-research setting.

(2) Deriving minimax rate for their setting. And the comparison of privacy bounds against nonprivacy bounds is very informative.

This work could point a future direction for many subsequent works.

2. Clarity: This paper is very easy to read, even with a large number of maths notations and equations. Explanation of each theorem is clear and thorough.

3. Quality: a very high quality paper.

4. Minor point: this paper does not contain any experiment, even not on simulated dataset. It would be interesting to see the actual performance of the proposed algorithm. And even some toy examples which serves for confirming the conclusions in the theorems would be interesting and worthwhile to include (maybe in the appendix).

**Time Spent Reviewing:**

2 hours

---

> ### Author Response · Authors · 2021-08-10
> **Response to the comment on numerical results.**
>
> Thank you very much for your appreciation and constructive comments.  We agree that some numerical examples would be necessary and we shall add them in the revision.

---

> > ### Comment · Reviewer_ALYo · 2021-08-31
> > **response**
> >
> > Thank you for your reply and I will take it into my consideration.

---

### Official Review · Reviewer_dv2E · 2021-07-16

**Rating:** 3
**Confidence:** 4

**Summary:**

This paper studies the change point detection problems in the multivariate nonparametric regression under the constraint of local differential privacy. Specifically, the goal is to detect changes in the regression function as soon as the change occurs. To achieve LDP, the proposed method first uses a locally private mechanism to transform the data point (x,y) to another random pair and then uses a classical change point detection algorithm with newly defined CUSUM statistics to detect the change. Theoretical false alarm probability analysis and the minimax detection delay properties are given. To quantify the cost of privacy, the paper also provides the optimal rate in the non-private setting.

**Limitations And Societal Impact:**

Some discussion of the limitation is given in the conclusion section. I don't see any negative societal impact of this paper.

**Main Review:**

I have a main correctness concern on the false alarm constraint. It is well-known that any change point detection algorithms will stop sooner or later. See some classical papers in change point detection literature [Page 1954; Shiryaev 1963; Lorden 1971]. Intuitively, at each time, there's a small error probability, and it will accumulate over time. So the classical false alarm constraint is on the average run length: E(\hat{t}|\Delta=\infty)\ge \gamma. It's not clear to me how you can get equations (5) and (10) that P_\infty(t<\infty)<\gamma. And this false alarm probability also appears as the main constraint in Theorem 2 and 4. I would suggest the authors give more explanation on that.

It'd be helpful if the authors can elaborate more on Definition 1. I'm confused about why it is a CUSUM statistic. My understanding is CUSUM stands for cumulative sum, and it is usually based on likelihood ratio statistics.

It's also not clear how the signal-to-noise ratio is defined. Assumption 4 is hard to parse.

Additionally, I don't see any fundamental differences in detecting the change in data vs. detecting the change in the regression parameters. Some motivating examples or explanations would be helpful.

Finally, I would suggest the authors add some experimental results to empirically demonstrate the usefulness of the proposed method.

**Time Spent Reviewing:**

6

---

> ### Author Response · Authors · 2021-08-10
> **Responses to the comments on the several change point detection aspects.**
>
> Thank you for your interests and constructive comments. In the following, we response to your comments point-by-point.
>
> * __On the false alarm probability.__  You are indeed right that the average run length control is widely used in the online change point detection literature as a false alarm control.  In this paper, we adopt the overall Type-I error control, which can be seen as an alternative and which is obtaining increasing popularity in the recent literature.   We now explain __why we prefer the overall Type-I error control__.  In theory, if one adopts the average run length control, then one should choose $\gamma$ properly with some prior knowledge on where the true change point is, if it exists.  To be specific, $\gamma$ should be larger, but not by too much, than $\Delta$ - the location of the change point; otherwise, the detection delay lacks a desirable upper bound.  More detailed discussions on this front can be found in https://arxiv.org/abs/2006.03283.  __On a technical level__, as you say, at each time $t$ there is a small error probability, but since our thresholds $b_{s,t}$ depend on $t$, this error probability also depends on $t$. Our thresholds are chosen in such a way that the infinite sum of these error probabilities is bounded by $\gamma$.  This is done by a union bound argument and the fact that $\sum_{t = 1}^{\infty} t^{-1}(t+1)^{-1} = 1$.  We will add further explanations of this in any revision.
>
> * __On Definition 1.__  It is a normalised difference between the estimators obtained from two segments of data.  We adopt the term CUSUM following the current trend in the literature and we are happy to call it differently.
>
> * __On the signal-to-noise ratio condition.__  To understand Assumption 4 better, we start with Assumption 5, which reads as
> 	$$\frac{\mbox{jump} \text{ size}^2 \times \mbox{change point location}}{\mbox{variance}} \times \mbox{width}^{\mbox{dimension}} \gtrsim \mbox{a logarithmic factor}.$$
> 	This is the signal-to-noise ratio condition required by the non-private counterpart result.  In fact, this is the nonparametric version of the univariate mean change point detection signal-to-noise ratio condition that
> 	$$\frac{\mbox{jump size}^2 \times \mbox{change point location}}{\mbox{variance}} \gtrsim \mbox{a logarithmic factor}.$$
> 	Comparing the above two equation displays, we see the only difference is the term $\mbox{width}^{\mbox{dimension}}$, which is due to the nonparametric nature of the problem.  We then investigate Assumption 4, which reads as
> 	$$\frac{\mbox{jump size}^2 \times \mbox{change point location}}{\mbox{variance} \vee M_0^2} \times \mbox{width}^{2 \times \mbox{dimension}} \times \alpha^2 \gtrsim \mbox{a logarithmic factor}.$$
> 	Comparing Assumptions 4 and 5, we see that there are three differences:
> 	(1). the privacy parameter $\alpha^2$;
> 	(2). $\mbox{width}^{2 \times \mbox{dimension}}$ instead of $\mbox{width}^{\mbox{dimension}}$; and
> 	(3). variance $\vee M_0^2$ instead of variance in the denominator.
> 	The points (1) and (2) are both due to the privacy constraint and (3) is due to the truncation, which is again introduced due to the privacy constraint.
>
> * __On the motivating example.__  You are indeed right that the changes in the regression parameters imply the changes in the data.  Having said this, knowing the changes lie in the regression functions means we have more information.  With this piece of information, detecting changes in regression functions is more efficient than detection changes in data.  Analogously, for a two-sample testing problem, if one knows that the potential distributional difference lies in the regression functions, then a test based on the regression functions is more powerful than a nonparametric test on the general distributions. In a public health setting (e.g. pubmed.ncbi.nlm.nih.gov/22759619), $X$ may represent demographic information, which is unlikely to change abruptly, while $Y$ may represent the prevalence of a disease, which could change quickly. Detecting changes in regression functions is likely to be more efficient in such settings.
>
> * __On the numerical experiments.__ We agree that some numerical examples would be necessary and we shall add them in the revision.

---

### Decision · Program_Chairs · 2021-09-28

**Decision:**

Accept (Poster)

**Comment:**

This paper studies the change point detection under the constraint of LDP. Specifically, at each time point t, a new user arrives with data of the form (X_t, Y_t), where X_t is a feature vector and Y_t is a response variable. The goal is to detect changes in the regression function as soon as the change occurs. To achieve LDP, the proposed method first uses a locally private mechanism to transform the data point (x,y) to another random pair and then uses an existing change point detection algorithm to detect the change. The reviewers agree that while this is a solid paper, it just misses the bar for NeurIPS.

**Consistency Experiment:**

NeurIPS has a long history of experimentation. In 2014, NeurIPS ran an experiment in which 10% of submissions were reviewed by two independent committees to quantify the randomness in the review process. This year, we repeated a variant of this experiment to see how the quality of the review process has changed over time.  This paper was part of the experiment and was therefore assigned to two committees (consisting of reviewers, an Area Chair, and a Senior Area Chair) that reached independent decisions.  If both committees made the same recommendation, this recommendation was followed. If a single committee recommended acceptance, the paper was accepted (with the exception of a few cases in which the other committee identified what we considered a fatal flaw, e.g., an error in a key result).

This copy’s committee reached the following decision: **Reject**

The other committee assigned to the paper recommended **Accept (Poster)**.  You can find the other set of reviews, along with any follow up discussion with the authors here:
https://openreview.net/forum?id=9GYcNKOuF4V